



**Variability of Lidar-Derived Particle Properties Over West Africa Due to Changes in**
**Absorption: Towards an Understanding**
Igor Veselovskii[1], Qiaoyun Hu[2], Philippe Goloub[2], Thierry Podvin[2], Mikhael Korenskiy[1],
Yevgeny Derimian[2], Michel Legrand[2], Patricia Castellanos[3]
*[1]General Physics Institute, Moscow, Russia.*
*[2]Univ. Lille, CNRS, UMR 8518 - LOA - Laboratoire d'Optique Atmospherique, Lille F-59000,*
*France*
*[3]NASA Goddard Space Flight Center, Greenbelt, USA*
**Abstract**
Measurements performed in Western Africa (Senegal) during the SHADOW-2 field
campaign are analyzed to show that spectral dependence of the imaginary part of the complex
refractive index (CRI) of dust can be revealed by lidar-measured particle parameters. Observations
in April 2015 provide good opportunity for such study, because, due to high optical depth of the
dust, exceeding 0.5, the extinction coefficient could be derived from lidar measurements with high
accuracy and contribution of other aerosol types, such as biomass burning, was negligible. For
instance, in the second half of April 2015, AERONET observations demonstrated a temporal
decrease of the imaginary part of CRI at 440 nm from approximately 0.0045 to 0.0025. This
decrease is in line with a change in relationship between lidar ratios (the extinction-to-
backscattering ratio) at 355 nm and 532 nm ($S_{355}$ and $S_{532}$). In the first half of April, $S_{355}/S_{532}$ is as
high as 1.5 and the backscatter Angstrom exponent $A_\beta$, is as low as -0.75, while after 15 April
$S_{355}/S_{532}=1.0$ and $A_\beta$ is close to zero. The aerosol depolarization ratio $\delta_{532}$ for the whole April
exceeded 30% in the height range considered, implying that no other aerosol, except dust,
occurred. The performed modeling confirmed that the observed $S_{355}/S_{532}$ and $A_\beta$ values match the
spectrally dependent imaginary part of the refractive index as can be expected for mineral dust
containing iron oxides. West Africa is also known for significant biomass burning aerosol
emissions during the dry season in the Sahel region.
The second phase of the SHADOW-2 campaign was focused on evaluation of lidar ratio
of smoke and estimates of its dependence on relative humidity (RH). For considered five smoke
episodes the lidar ratio increases from 44±5 sr to 66±7 sr at 532 nm and from 62±6 sr to 80±8 sr
at 355 nm, when RH varied from 25% to 85%. Performed numerical simulations demonstrate, that
observed ratio $S_{355}/S_{532}$, exceeding 1.0 in the smoke plumes, can indicate to increase of the
imaginary part of the smoke particles in UV.


## 1. Introduction

Atmospheric dust provides significant impacts on the Earth's climate system and this
impact remains highly uncertain (IPCC report, 2013). In modeling the direct aerosol effect, the
vertical profile of aerosol extinction is one of the basic input parameters, and when this profile is
derived from the elastic backscatter lidar observations, the knowledge of the extinction-to-
backscatter ratio (so called lidar ratio) is essential. Although the desert dust in source regions is
sometimes qualified as "pure dust", it is always a mixture of various elements, e.g. iron oxides,
clays, quartz and calcium–rich species, which proportions can vary (Sokolik and Toon, 1999;
Wagner et al., 2012; Di Biagio et al., 2017, 2019 and references therein). Thus, the dust optical
properties, and hence the lidar ratio (S) can vary, depending on relative abundance of various
minerals in emission sources. Imaginary part of the complex refractive index (CRI) of different
minerals can vary spectrally and often exhibits an increase in UV for dust, containing iron oxides.
Therefore, retrieval of the dust extinction profiles from elastic backscatter lidar observation should
account for the spectral variation of the lidar ratio.
The Raman and HSRL lidars are capable to provide independent profiling of aerosol
backscattering and extinction coefficients (Ansmann et al., 1992), and therefore are widely used
to measure the lidar ratios of dust from different origins (e.g. Sakai et al., 2003; Papayannis et al.,
2008, 2012; Xie et al., 2008; Ansmann et al., 2011; Mamouri et al., 2013; Burton et al., 2014;
Nisantzi et al., 2015; Giannakaki et al., 2016; Hofer et al., 2017, 2019). The African deserts are
the largest sources of mineral dust and numerous studies have been conducted for quantifying the
particle intensive parameters (parameters independent of concentration) during dust transport from
this source region to Europe and over the Atlantic Ocean (Mattis et al., 2002; Amiridis et al., 2005;
Mona et al., 2006; Papayannis et al., 2008; Preißler et al., 2013; Rittmeister et al., 2017). The dust
properties are, however, modified during the transport, experiencing mixing and aging processes,
thus characterization of dust properties near the source regions is highly important for evaluation
the parameters of "pure dust". The lidar ratios at 355 nm and 532 nm ($S_{355}$ and $S_{532}$) were measured
during the SAMUM-1 and 2 experiments in Morocco and Capo Verde respectively (Esselborn et
al., 2009; Tesche et al., 2009, 2011; Groß et al., 2011; Ansmann et al., 2011), as well as during the
more recent SHADOW-2 experiment in Senegal (Veselovskii et al., 2016, 2018). The lidar ratios
$S_{355}$ and $S_{532}$ measured during SAMUM experiments didn't present significant spectral
dependence. For example, for SAMUM-2 campaign, the averaged values of $S_{355}$ and $S_{532}$ are
53±10 sr and 54±10 sr respectively (Tesche et al., 2011). During SHADOW, however, $S_{355}$
significantly exceeded $S_{532}$ in many dust episodes, which was linked to an increase of the
imaginary part of CRI of dust at 355 nm (Veselovskii et al., 2016).





The dust backscattering coefficient (and so lidar ratio), in contrast to extinction coefficient,
is sensitive to the imaginary part of CRI (Perrone et al., 2004; Gasteiger et al., 2011). Thus, it is
expected that enhanced absorption in the UV should increase the lidar ratio. In turn, the ratio
$S_{355}/S_{532}$ should characterize the spectral variation of the imaginary part of CRI. The latest version
of AERONET products (3.0) provides inversions of lidar related properties, including the lidar
ratio, from almucantar scans with ground-based sun photometers. For these products, the shortest
available wavelength is 440 nm. Despite $Im_{440}$ is lower than $Im_{355}$, AERONET observations still
show an increase of absorption at 440 nm in respect to 675 nm that yields a ratio of $S_{440}/S_{675}$ close
to 1.4 for Saharan dust (Shin et al., 2018). The goal of this work is to analyze the correlation of
variations of $Im_{440}$ from AERONET with measured values from lidar to reveal the effect of dust
absorption on lidar-derived aerosol properties. We focus on height and day-to-day variations of
the dust intensive properties, such as $S_{355}$ and $S_{532}$, depolarization ratio ($\delta$), as well as the extinction
and backscatter Ångström exponents ($A_\alpha$ and $A_\beta$ respectively) measured during several strong dust
episodes in April 2015 during the SHADOW-2 campaign.
The smoke aerosol particles, typically originated from biomass burning, can also have a
pronounced spectral dependence of absorption. This is generally due to presence of carbonaceous
particles with organic compounds, so-called brown carbon (BrC) (Sun et al., 2007; Kirchstetter, et
al., 2004). The Sahel region is known for seasonal biomass burning caused by human activity on
combustion of agricultural waste that can produce an abundant amount of BrC. The smoke can
also be mixed with mineral dust during long-range transport or in the emission origin (Haywood
et al., 2008). During the SHADOW-2 the observation period included the biomass burning season,
thus an additional effort was dedicated to examination of spectral lidar ratio variability of
transported biomass burning aerosol under different environmental conditions and presents a
supplementary subject of the current study.
The paper is organized as follows. Section 2 describes the lidar system and provides the
main expressions used for the data analysis. Several strong dust episodes, in April 2015, are
analyzed in Section 3. In Section 4, the smoke episodes occurring from December 2015 to January
2016, are used to evaluate the variation of the smoke lidar ratio with relative humidity. The paper
is finalized with conclusion.

**2.    Experimental setup and data analysis**
The observations were performed with LILAS multiwavelength Raman lidar during
SHADOW-2 campaign at Mbour, Senegal. Information related to the SHADOW-2 and
observation site is presented in Veselovskii et al. (2016). The LILAS is based on a tripled Nd:YAG
laser with a 20 Hz repetition rate and pulse energy of 90/100/100 mJ at 355/532/1064 nm. The



aperture of the receiving telescope is 400 mm. During the campaign, LILAS configuration
$(3\beta+2\alpha+1\delta)$ allowed the measurement of three particle backscattering ($\beta_{355}$, $\beta_{532}$, $\beta_{1064}$), two
extinction coefficients ($\alpha_{355}$, $\alpha_{532}$) and depolarization ratio at 532 nm ($\delta_{532}$). To improve the
performance of the system at 532 nm the rotational Raman channel was used instead of the
vibrational one (Veselovskii et al, 2015). The measurements were performed at a 47 degrees angle
to horizon. The backscattering coefficients and depolarization ratios were calculated with a 7.5 m
range resolution (corresponding to 5.5 m vertical resolution), while range resolution of extinction
coefficient varied from 50 m (at 1000 m) to 125 m (at 7000 m). Particle extinction and
backscattering coefficients at 355 nm and 532 nm are calculated from elastic and Raman
backscatter signals, as described in Ansmann et al. (1992). An additional Raman reception channel
at 408 nm was setup for profiling the water vapor mixing ratio (WVMR) (Whiteman et al., 1992).

The particle depolarization ratio $\delta$, determined as a ratio of cross- and co-polarized

components of the particle backscattering coefficient, was calculated and calibrated the same way
as described in Freudenthaler et al. (2009). To further the analysis of complex aerosol mixtures,
containing dust (d) and smoke (s), we can write $\beta = \beta^d + \beta^s$ and $\alpha = \alpha^d + \alpha^s$. The depolarization
ratio of such a mixture is therefore:
$$\delta = \frac{\left(\frac{\delta^d}{1+\delta^d}\right)\beta^d + \left(\frac{\delta^s}{1+\delta^s}\right)\beta^s}{\frac{\beta^d}{1+\delta^d} + \frac{\beta^s}{1+\delta^s}} \qquad (1)$$
Here $\delta^d$ and $\delta^s$ are the particle depolarization ratios of dust and smoke components respectively.

To characterize the spectral dependence of the extinction ($\alpha$) and backscattering ($\beta$)

coefficients, corresponding Ångström exponents are introduced as:
$$A_\alpha = \frac{\ln\left(\frac{\alpha_{\lambda_1}}{\alpha_{\lambda_2}}\right)}{\ln\left(\frac{\lambda_2}{\lambda_1}\right)} \text{ and } A_\beta = \frac{\ln\left(\frac{\beta_{\lambda_1}}{\beta_{\lambda_2}}\right)}{\ln\left(\frac{\lambda_2}{\lambda_1}\right)} \qquad (2)$$
Where $\alpha_{\lambda_1}$, $\alpha_{\lambda_2}$, $\beta_{\lambda_1}$, $\beta_{\lambda_2}$ are the extinction and backscattering coefficients at wavelengths $\lambda_1$ and
$\lambda_2$. For the mixture of smoke and dust, the extinction Ångström exponent (EAE) can be calculated
from the ratio $\frac{\alpha_{\lambda_1}}{\alpha_{\lambda_2}}$:



$$\frac{\alpha_{\lambda_1}}{\alpha_{\lambda_2}} = \frac{\alpha_{\lambda_1}^d + \alpha_{\lambda_1}^s}{\alpha_{\lambda_2}^d + \alpha_{\lambda_2}^s} = \frac{\alpha_{\lambda_1}^d}{\alpha_{\lambda_2}^d}\frac{\left(1+\frac{\alpha_{\lambda_1}^s}{\alpha_{\lambda_1}^d}\right)}{\left(1+\frac{\alpha_{\lambda_2}^s}{\alpha_{\lambda_2}^d}\right)} = \frac{\alpha_{\lambda_1}^d}{\alpha_{\lambda_2}^d}\frac{\left(1+\frac{\alpha_{\lambda_2}^s(\frac{\lambda_2}{\lambda_1})^{A_\alpha^s}}{\alpha_{\lambda_2}^d(\frac{\lambda_2}{\lambda_1})^{A_\alpha^d}}\right)}{\left(1+\frac{\alpha_{\lambda_2}^s}{\alpha_{\lambda_2}^d}\right)} = \frac{\alpha_{\lambda_1}^d}{\alpha_{\lambda_2}^d}\frac{\left(1+\frac{\alpha_{\lambda_2}^s}{\alpha_{\lambda_2}^d}\left(\frac{\lambda_2}{\lambda_1}\right)^{(A_\alpha^s - A_\alpha^d)}\right)}{\left(1+\frac{\alpha_{\lambda_2}^s}{\alpha_{\lambda_2}^d}\right)} \tag{3}$$

Here $A_\alpha^d$ and $A_\alpha^s$ are the extinction Ångström exponents of dust and smoke. The Ångström
exponent of the mixture is obtained from (3):
$$A_\alpha = \frac{\ln\frac{\alpha_{\lambda_1}}{\alpha_{\lambda_2}}}{\ln\frac{\lambda_2}{\lambda_1}} = A_\alpha^d + \frac{1}{\ln\frac{\lambda_2}{\lambda_1}}\ln\left[\frac{\left(1+\frac{\alpha_{\lambda_2}^s}{\alpha_{\lambda_2}^d}\left(\frac{\lambda_2}{\lambda_1}\right)^{(A_\alpha^s - A_\alpha^d)}\right)}{\left(1+\frac{\alpha_{\lambda_2}^s}{\alpha_{\lambda_2}^d}\right)}\right] \tag{4}$$

The backscattering Ångström exponent (BAE) can be calculated in a similar way. And finally, the
lidar ratio of the aerosol mixture is calculated as:
$$S = \frac{S^d\beta^d + S^s\beta^s}{\beta^d + \beta^s} = S^d + \frac{\beta^s}{\beta}(S^s - S^d) \tag{5}$$

where $S^d$ and $S^s$ are the lidar ratios of dust and smoke.

**3. Dust observations in March and April 2015**

The aerosol over West Africa presents strong seasonal variations. The spring is

characterized by strong dust emission, while, during winter season, intense forest fires occurring
in the equatorial regions emit smoke particles that are transported over Senegal. The SHADOW-
2 campaign included the following periods of measurements: 13 March – 25 April 2015, 8–25
December 2015 and 5-24 January 2016, so numerous dust and smoke episodes were observed. In
our analysis of lidar-derived aerosol properties, we considered also aerosol columnar properties
provided by AERONET (Holben et al. 1998) and aerosol profiles predicted by the Modern-Era
Retrospective analysis for Research and Applications, Version 2 (MERRA-2) aerosol reanalysis
(Gelaro et al., 2017; Randles et al., 2017). MERRA-2 is the first long-term global reanalysis to
assimilate space-based aerosol observations and include their radiative coupling with atmospheric
dynamics. MERRA-2 is driven by the Goddard Earth Observing System (GEOS) model version 5
that includes the Goddard Chemistry, Aerosol, Radiation and Transport (GOCART) module.
GOCART models the sources, sinks, and transformation of the following five aerosol species as
external mixtures: dust, organic carbon (OC), black carbon (BC), sulfates (SU) and sea salt (SS).
Dust and sea salt are represented by five non-interacting size bins, and have wind-speed dependent





emissions. The MERRA-2 reanalysis assimilates AOD observations from the twin Moderate
Resolution Imaging Spectroradiometer (MODIS) instruments, MODIS-Terra and MODIS-Aqua,
as well as the AERONET ground-based sun photometer network. In addition, the profiles of
meteorological variables (P, T, RH), provided by radio-sondes at the Dakar airport, located ~70
km from the M'bour site, were also available. The relative humidity (RH) profiles over the M'bour
site were calculated from the combination of lidar-derived WVMR and temperature profile from
radiosounding.
Fig.1 shows the aerosol optical depth at 532 nm ($AOD_{532}$) for March, April and December
2015 recalculated from AERONET AOD at 500 nm using 440-675 nm Ångström exponent. The
same figure shows the AODs for the five aerosol species used in MERRA-2 model, such as dust,
organic carbon (OC), black carbon (BC), sulfates (SU) and sea salt (SS). The optical depths
provided by MERRA-2 and AERONET are in a good agreement. Dust is the predominant aerosol
component for all three months with the highest values of AOD in April. The contribution of
organic carbon (the main component of the biomass burning products) is significant in December,
when the forest fire season starts in equatorial regions, though noticeable amount of OC is
predicted also for March and for the beginning of April. The contribution of BC and SU to the
total AOD is low: the sum of the corresponding AODs is below 0.1 for all three months.
The single scattering albedo (SSA) over the M'Bour site in 2015 provided by AERONET
at 440 and 675 nm is shown in Fig.2. The $SSA_{675}$ is above 0.97 for March – April period, but at
440 nm dust absorption is stronger and, in March, $SSA_{440}$ is about 0.9. However, in the middle of
April, $SSA_{440}$ increases up to 0.95, indicating that aerosol becomes less absorbing at shorter
wavelengths. We can thus expect that variation of SSA at 355 nm between April and March should
be even stronger. In our study we consider two groups of observation. The first group corresponds
to the beginning of April, when SSA at 440 nm was lower. The second group covers the second
half of April, when SSA at 440 nm increased. By analyzing these two groups we expect to reveal
the effect of aerosol absorption, on lidar-derived aerosol properties.

### 182 3.1. Dust episode on 1 – 4 April 2015

In the beginning of April the dust was transported by Continental trades (Harmattan) from
the northeastern/eastern drylands. For period 1 - 4 April, as follows from Fig.1b, the $AOD_{532}$ over
Dakar increased up to 1.0. Fig.3 shows spatio-temporal distributions of the aerosol backscattering
coefficient $\beta_{532}$, particle depolarization ratio $\delta_{532}$, and water vapor mixing ratio for the nights 1-2,
2-3 and 3-4 April 2015. Corresponding back-trajectories, shown in Fig.4, demonstrate that, on 1-
2 and 2-3 April, air masses at all heights arrive from the North-East, whereas on 3-4 April the air
masses above 2500 m are advected from the East. These air masses are characterized by higher





humidity and may contain biomass-burning products. During these three nights, depolarization
ratio and WVMR present some evolution. On 1-2 April $\delta_{532}$ exceeds 30% and does not change
significantly within the dust layer, even if some decrease is observed above 2000 m after 03:00
UTC. By 3-4 April the depolarization ratio above 2500 m decreases below 25%, simultaneously
with increase of the WVMR. During the dust episode, the relative humidity did not exceed 20%
on 1-3 April, but on 3-4 April it increased up to 40% above 2500 m.

Vertical profiles of dust particle properties such as aerosol extinction coefficients $\alpha_{355}$, $\alpha_{532}$,

particle depolarization ratio $\delta_{532}$ and lidar ratios $S_{355}$, $S_{532}$ are shown on Fig.5 for the three
observation periods on 1, 2-3 and 3-4 April 2015. The corresponding extinction and backscatter
Ångström exponents, calculated for 355 and 532 nm wavelengths, are presented in Fig.6. During
all three observation periods $A_\alpha$ is slightly negative ($A_\alpha$ = -0.1±0.1) up to 2000 m. For the dust
component, MERRA-2 provides value of $A_\alpha$ =-0.14, which agrees with observations. Above 2000
m, $A_\alpha$ exhibits some increase, which is most significant on 3-4 April, when $A_\alpha$ reaches 0.3±0.1 at
4000 m height. Simultaneous decrease of $\delta_{532}$ indicates to the possible presence of smoke particles
above 2000 m. The backscatter Ångström exponent $A_\beta$, in contrast with $A_\alpha$, is sensitive to the
spectral dependence of the imaginary part of CRI, thus yielding complicated vertical variability of
$A_\beta$. In particular, on 2-3 April $A_\beta$ decreases from -0.5 to -0.7 within 1500–2500 m height range,
when $A_\alpha$ remains stable.

As follows from Fig.5, on 1 April the lidar ratio $S_{355}$=70±6 sr does not change with height,

while $S_{532}$ gradually decreases from 60±5 sr at 1000 m to 50±4 sr at 3000 m height. On sessions
that followed (Fig.5b,c) the lidar ratios at both 355 nm and 532 nm decreased. Thus, the range of
lidar ratios variation for the dust episode on 1-4 April is 60-70 sr at 355 nm and 45-60 sr at 532
nm. The lidar ratios ($S_{355}$ and $S_{532}$) modeled by MERRA-2 for the dust component are also shown
on Fig. 5. Corresponding values are of 70 sr and 42 sr respectively and do not vary with altitude
as the model optical properties of all dust size bins based on spectral complex refractive indices
from the Optical Properties of Aerosols and Clouds (OPAC) tables (Hess et al. 1998) and the
spheroidal shape models developed by Meng et al. (2010) are the same and fixed, as dust is treated
as homophobic. Modeled value $S_{355}$ is near the top of the range of observed values, while modeled
$S_{532}$ underestimates the observations.

The gradual decrease of $S_{532}$ with height in Fig.5a,c is however unusual. There are, at least,

two possible reasons to explain $S_{532}$ height variation. The first one can be the presence of non-dust
particles, for example, smoke. The second reason is that the properties (composition) of dust
change with height. If non-dust particles are present, the particle intensive properties, such as S, $\delta$
and $A_\alpha$ should vary with height in consistent way. The MERRA-2 modeling reported in Fig.1





shows that in the beginning of April the organic carbon is the second main contributor to the AOD,
after dust. We should recall, however, that the model can provide a realistic range of OC variation,
however not necessarily reproducing the exact spatio-temporal distribution of OC extinction
coefficient.

In the dust episode considered, the most significant smoke contribution was observed on

3-4 April. Fig.7a shows the profiles of measured $\alpha_{355}$ and $\alpha_{532}$ together with MERRA-2 modeled
extinction coefficients at 532 nm for five aerosol components. The extinction Ångström exponents
measured by lidar and modeled by MERRA-2 for dust component are given by Fig.7b. The same
figure shows also the lidar derived water vapor mixing ratio profile together with the relative
humidity. At low altitudes (below 2500 m), where aerosol is represented by pure dust, the
measured and modeled values of extinction coefficients are close. Above 2500 m the measured
value of $\alpha_{355}$ exceeds that of $\alpha_{532}$, indicating the presence of smoke particles, while modeled
contribution of OC to the total extinction is very low. The measured extinction Ångström exponent
is about -0.1 below 2000 m, which well agrees with modeling results for pure dust. Increase of
WVMR and RH above 2000 m coincides with growth of the $A_\alpha$. For the considered case, the
model reproduces correctly the dust loading, but underestimates the smoke contribution. At 3500
m, the difference between measured and modeled $\alpha_{532}$ is about 0.045 km$^{-1}$ which can be attributed
to the smoke contribution.

Dust and smoke particles contributions to the total backscattering coefficient can be also

separated on the basis of the depolarization measurements, assuming that depolarization ratios of
these particles are known (Tesche et al., 2009). The results of such decomposition are presented
in Fig.7c, assuming 35% and 7% for dust and smoke depolarization ratio, respectively. The
contribution of smoke to the total $\beta_{532}$ at 3500 m is 0.0009 km$^{-1}$sr$^{-1}$. For the smoke lidar ratio of 50
sr at 532 nm (validity of this choice will be discussed in section 3.3), the smoke extinction
coefficient is about 0.045 km$^{-1}$. This value agrees well with smoke contribution obtained from
Fig.7a at 3500 m and thus can be used for estimating the smoke effect on the intensive aerosols
properties derived from lidar measurements.

The depolarization ratio of the "dust-smoke" mixture, calculated with expression (1),

matches the observed value since decomposition in Fig.7c is based on depolarization
measurements. The Ångström exponent at 3500 m computed with (4) for $\alpha_{532}^s$ =0.045 km$^{-1}$, $\alpha_{532}^d$
=0.147 km$^{-1}$, $A_\alpha^d$ =-0.1 and $A_\alpha^s$ =1.0 yields $A_\alpha$ =0.2, which matches observed value 0.25±0.1.
Hence, the observed variation of $A_\alpha$ above 2000 m on 3-4 April is explained by smoke
contribution. In a similar way, using (5) we can estimate the smoke lidar ratio ($S_{532}^s$) that would





match the observed decrease of $S_{532}$. To explain decrease of the lidar ratio at 3500 m from 50 sr
to 45 sr, the smoke lidar ratio should be about 25 sr, which is unrealistically small. Such small
lidar ratio could be attributed to the maritime aerosol, but then the lidar ratios at both wavelengths
should decrease simultaneously. Recall that on 1-2 April smoke contribution was significantly
lower, while decrease of $S_{532}$ is about 10 sr. Thus, smoke particles presence cannot explain the
observed decrease of $S_{532}$ and it should be probably attributed to changes of dust composition (and
so the imaginary part) with height.
Smoke lidar ratio is usually assumed to be higher than that of dust (Burton et all., 2014),
meanwhile in Fig 5c the lidar ratio $S_{532}$ is not increased in presence of the smoke particles. It should
however be noticed that our results were obtained at low RH. The smoke particles are hygroscopic
and the lidar ratio should increase with RH. The way to characterize $S_{532}^s$ over Dakar site can be
based on the analysis of the lidar measurements during smoke episodes within height range where
smoke contribution becomes predominant. The results of such analysis will be discussed later in
section 3.3.

**3.2. Dust episodes on 14 and 24 April 2015**.
In the second part of April 2015, dust $AOD_{532}$ exceeded 1.0 (Fig.1b) and contributions of
other aerosol components were insignificant. Meanwhile, as follows from Fig.2, $SSA_{440}$ increased
after 15 April, thus dust became less absorbing in UV, which should influence the lidar-derived
aerosol intensive properties. Fig.8 shows extinction coefficients and lidar ratios at 355 nm and 532
nm, together with depolarization ratio $\delta_{532}$ and the Ångström exponents $A_\alpha$ and $A_\beta$ observed on
14 April (00:00 – 05:00 UTC) and 23-24 April (23:00-06:00 UTC). The first case is a "transition
day" when $SSA_{440}$ starts to increase. Extinction profiles presented in Fig.8a show that two dust
layers can be distinguished.  In the first layer (below 2.5 km), aerosol intensive properties are
similar to that of 1-4 April with $S_{355}>S_{532}$, slightly negative $A_\alpha = -0.1$ and $A_\beta$ as low as -0.35. In
the second layer $S_{355}$ and $S_{532}$ coincide and both $A_\alpha$ and $A_\beta$ are close to zero. The depolarization
ratio in the second layer is about 31%, slightly lower than in the first one. Thus, we can assume
that increase of the imaginary part in UV in the first layer is more significant, than in the second
one. From back-trajectories given in Fig.9, we can conclude that the air masses in the first layer
originate from the Northeastern/Eastern drylands, while in the second layer the air masses arrive
from the East. After 14 April, $S_{355}$ and $S_{532}$ coincided for the whole height range and results
obtained on 23-24 April (Fig.8 c, d) are the example of such observations. Back-trajectories show
that the air masses at both 2.0 and 3.0 km height are transported from East. The ratio $S_{355}/S_{532}$ is





close to 1.0 within the whole dust layer and both Ångström exponents $A_\alpha$, $A_\beta$ are close to zero.
Thus, the results from Fig.8, 9 are indicating that lidar-derived aerosol properties depend on the
dust source origin.

**3.3 Analysis of lidar ratio variations in March – April 2015**
Fig.10 summarizes the lidar ratio measurements for period from 29 March to 24 April 2015
(first phase of SHADOW ended on 25 April). Here we focus on the properties of the "pure dust",
thus do not show results before 29 March, when AOD was lower and the contribution of other
aerosol types could be significant (Fig.1). For the Fig.10 we have chosen height intervals, where
S value is stable and δ exceeds 30%. For example, on 14 and 24 April lidar ratios are averaged
inside 2.7-3.7 km and 2.0-4.0 km layers respectively. For the period considered, $S_{355}$ and $S_{532}$ vary
in the ranges 50 sr – 80 sr and 45 sr - 60 sr respectively with a mean values of 62 sr and 51 sr.
Enhanced variability of $S_{355}$ compared to $S_{532}$ can be explained by variation of the imaginary part
at 355 nm. At the beginning of the 29 March and 8 April dust episodes, $S_{355}/S_{532}$ ratio is as high as
1.5 and then gradually decreases. After 14 April, $S_{355}/S_{532}$ ratio becomes close to 1.0, thus S
presents no spectral dependence.
The day-to-day variation of aerosol column properties, including the spectrally dependent
complex refractive index, can be obtained from AERONET (Holben et al., 1998). Fig.11 shows
the imaginary part of the refractive index at 440 nm and 675 nm ($Im_{440}$, $Im_{675}$) provided by
AERONET for the same period of time as in Fig.10. The $Im_{440}$ strongly decreases after 14 April,
correlating with the decrease of $S_{355}/S_{532}$ ratio in Fig.10, which corroborates the suggestion, that
variations of $S_{355}/S_{532}$ ratio are related to variation of dust absorption in UV. The retrieved real
part (Re) of the complex refractive index oscillates around Re=1.45 and shows no significant
spectral dependence. Correlation between enhancement of $Im_{440}$, with in respect to $Im_{675}$, and
increase of lidar-derived $S_{355}/S_{532}$ is clearly seen in Fig.12, showing time – series of difference
$Im_{440}-Im_{675}$ and $S_{355}/S_{532}$ ratio.
To analyze the variations of observed lidar ratios and the Ångström exponents, a simplified
numerical simulation has been performed. For a realistic modeling of the dust lidar ratio, various
mixtures of different mineral components and particles shapes should be considered. Sensitivity
of the modeling results to the dust mixture parameters was demonstrated in study of Gasteiger et
al. (2011). Such detailed modeling, however, is out of the scope of the present paper. Here we only
intend to evaluate the main impact when the imaginary part of CRI is modified.
The lidar ratio depends not only on the complex refractive index but also on the dust
particle size distribution (PSD). The PSDs provided by AERONET on 2 and 23 April 2015 (three





distributions for each day) are shown in Fig.13. The PSDs are similar and the effective radii for
both days are about 0.75 μm, thus, difference in S observed for 2 and 23 April should be related
mainly to the complex refractive index. Fig.14a presents modeled $S_{355}$ and $S_{532}$ lidar ratios together
with the extinction and backscattering Ångström exponents $A_\alpha$, $A_\beta$ as a function of the imaginary
part. Computations were performed for the AERONET derived size distribution on 23 April from
Fig.13 using the assembly of randomly oriented spheroids (Dubovik et al., 2006) with the real part
Re=1.55. $S_{355}$ and $S_{532}$ increase with the imaginary part and the ratio $S_{355}/S_{532}$ is about 1.1.
Extinction coefficient is slightly sensitive to the imaginary part, thus increase of S in Fig.14 is due
to decrease of backscattering coefficient with Im. The modeled $A_\alpha$ is about $A_\alpha$=0.1, while $A_\beta$
decreases with Im to $A_\beta$=-0.2. To estimate the influence of a spectrally dependent imaginary part
Im(λ) on $A_\beta$, we have also performed computations assuming a fixed $Im_{532}$=0.002 and only $Im_{355}$
is free to vary. Corresponding results are shown in Fig.14a with open stars. Spectral dependence
of the imaginary part significantly decreases $A_\beta$: for $Im_{355}$=0.005 ($Im_{355}$ − $Im_{532}$=0.003), $A_\beta$
decreases to -0.75.

We should recall however, that for the second half of April the observed ratio $S_{355}/S_{532}$,

was about 1.0, and both extinction and backscatter Ångström exponents were close to zero. To
figure out the kind of PSD that would reproduce those observations, we retrieved the PSD from
3β+2α measurements, as described in Veselovskii et al. (2002, 2010). For that purpose, data from
23-24 April (Fig.8), averaged within 2-3 km layer, were inverted and corresponding PSD is shown
in Fig.13 with red line. Inversion was performed for the assembly of randomly oriented spheroids,
in assumption of spectrally independent refractive index. Due to the limited number of input data
(five) we are able to reproduce only the main features of the PSD. The maximum of this lidar
derived PSD is shifted towards larger radii, with respect to the AERONET size distribution, but at
the same time, retrieved PSD contains significant contribution from the fine particles. The
simulation results for this lidar derived PSD, are given by Fig.14b. The lidar ratios $S_{355}$, $S_{532}$ for
all values of the imaginary part are close. The backscatter and extinction Ångström exponents are
close to zero, matching the observations of the second half of April 2015. Thus simulation results
demonstrate dependence on the PSD chosen, but in both cases these lead to the same conclusion:
observed low values of $A_\beta$ can not be reproduced without accounting for spectral dependence of
the imaginary part.

To compare computations and observations, information upon $Im_{355}$ and $Im_{532}$ values is

needed. The recently measured refractive indices of dust, sampled at different regions of Africa,
are presented by Di Biagio et al. (2019). In particular, for the countries located North and East of
Senegal, the imaginary parts at 370, 470, 520, 660 nm are of 0.0043, 0.0033, 0.0026, 0.0013 for





Mauritania and 0.0048, 0.0038, 0.0030, 0.0024 for Mali respectively. The highest values of lidar
ratios, observed in our measurements, are about 60 sr and 80 sr at 532 nm and 355 nm respectively.
Corresponding imaginary parts of CRI from Fig.14 can be estimated as $Im_{532}$=0.002-0.003 and
$Im_{355}$=0.005-0.006, which agrees with results presented by Di Biagio et al. (2019). Assuming
$Im_{355}$=0.005 and $Im_{532}$=0.002, the modeled ratio $S_{355}/S_{532}$ is about 1.44 and $A_\beta$ is about -0.75 for
both AERONET and lidar derived PSDs, which again reasonably agrees with observations. The
modeling performed is very simplified, still it confirms that observed values of $S_{355}/S_{532}$ ratio and
$A_\beta$ can be explained by the spectral dependence of the imaginary part of CRI.

Thus, based on our measurement results, two types of dust can be distinguished. The first

type has high $S_{355}/S_{532}$ ratio (up to 1.5). Such kind of dust is characterized by increase of the
imaginary part in UV and it was observed, for example, during 29 March and 10 April episodes.
For the second type, the ratio $S_{355}/S_{532} \approx 1.0$, so variation of the imaginary between 532 and 355 nm
wavelengths should be smaller than for the first type. Such dust was observed in the second half
of April 2015. Both types are characterized by high depolarization ratio, $\delta_{532}$, exceeding 30%, so
depolarization measurements at 532 nm are not capable to discriminate between these two types
of dust.

Difference in the observed dust properties is probably related to the mineralogical

characteristics in the source region. From the back-trajectories analysis presented in Figs. 4 and 9
one can suppose that the first type of dust was transported from the North–East, while the second
type from the East. In order to verify if a difference in the dust emission source region and transport
take place, we also analyzed the Infrared Difference Dust Index (IDDI) derived from the Meteosat
Second Generation (MSG) geostationary satellite imagery in thermal infrared (TIR). The IDDI is
developed by Legrand et al. (1985, 2001) originally for the Meteosat First Generation (MFG) and
is based on impact of airborne mineral dust on TIR radiation emitted by terrestrial surface. The
physical principle of the IDDI derivation is in thermal contrast between terrestrial surface and
atmosphere and the best sensitivity is found at around noon time when the surface temperature is
maximal (Legrand et al., 1988). The IDDI product shows that brightness temperature of terrestrial
surface observed by satellite can be reduced up to about 50°K in presence of airborne mineral dust,
while reduction by about 10°K already indicates a major dust event (Legrand et al., 2001). A direct
relationship between the IDDI and aerosol optical thickness in solar spectrum and visibility was
also found (Legrand et al., 2001). It should be mentioned here that the IDDI was initially developed
for MFG and the absolute consistence with the IDDI values from MSG should be examined due
to differences in spatial and spectral resolutions between two sensors. However, the physical
principles used for the IDDI determination are the same and a direct application of the MFG IDDI
algorithm to MSG was found as possible. Moreover, tests showed that the absolute values of IDDI





for a coincident overlapping period of MFG and MSG are very close. Nevertheless, employment
of the IDDI from MSG is indeed applicable for the required in the current analysis purpose of
solely dust spatial patterns detection.

The IDDI calculations, applied to the MSG images taken during the field campaign, clearly

show a major dust event in northern and central Africa. The elevated IDDI values over Senegal
are also visible. The IDDI images show distinguishable changes in the emission sources and
transport features during the different phases of the observations. For instance, Fig. 15 shows that
the dust emissions during the first phase of the event are originated in south Algeria, Mauritania
and Mali (examples of images from 29 and 30 March 2015). Weeks later, spatial patterns of the
elevated IDDI are shifted to south and show source regions in south of Niger (Fig.15c, d). Of
course, attribution of emission sources mineralogy to aerosol spectral absorption is a complex task
(Alfaro., et al 2004; Lafon et al., 2006; Di Biagio et al., 2017, 2019) and it is difficult to point to a
specific source that could clearly explain the observed in this study change in the aerosol absorbing
properties. However, the IDDI images clearly suggest a change in the dust transport regime that is
consistent with the change in the dust optical properties.

**4.  Smoke episodes in December 2015 – January 2016**

During the SHADOW campaign, we had several strong smoke episodes in December 2015

– January 2016, when air mass transported the products of biomass burning from the areas of
intensive forest fires in equatorial region. The relative humidity in the advected smoke layers
varied from episode to episode, allowing evaluation of the RH influence on the smoke lidar ratios
$S_{355}$, $S_{532}$. The spatio-temporal evolution of the particle backscattering coefficient and
depolarization ratio at 532 nm, during the 14-15 December 2015 smoke episode, is given in Fig.16.
The same figure shows also the water vapor mixing ratio, a convenient tracer to identify wet air
mass arrived from the equatorial region. The smoke particles are usually contained in elevated
layers, being mixed with dust (Veselovskii et al., 2018). The height ranges where the smoke
particles are predominant can be identified by low depolarization ratio and enhanced WVMR. For
event considered, the smoke particles are predominant above 1500 m after midnight.

The vertical profiles of $\alpha_{355}$, $\alpha_{532}$, $S_{355}$, $S_{532}$, $A_\alpha$, $A_\beta$ together with the water vapor mixing

ratio and the relative humidity, for 15 December (04:00 – 06:00 UTC), are shown in Fig.17. The
same figure presents decomposition of $\beta_{532}$ to the dust and smoke contributions, based on
depolarization measurements (Tesche et al., 2011). The smoke episodes are characterized by
different relative humidity within the elevated layer. On 15 December, RH is about 40% in the
1500 – 2100 m range and the ratio $\frac{\beta_{532}^{s}}{\beta_{532}}$ is about of 0.57 at 2000 m. The lidar ratio $S_{532}$ decreases



from 50 sr to 44 sr in 1000 m - 2000 m range, while $S_{355}$ rises from 58 sr to 67 sr, thus $S_{355}$
significantly exceeds $S_{532}$. We should recall that lidar ratios presented in Fig.17 are attributed to
dust- smoke mixture. In principle, we can estimate $S_{532}^s$ using Eq.5, because the ratio $\dfrac{\beta_{532}^s}{\beta_{532}}$ is
available. Corresponding $S_{532}^s$ profile obtained for assumed $S_{532}^d$ =50 sr is shown in Fig.17a (black
line). $S_{532}^s$ is about 40 sr at 2000 m and it is close to measured $S_{532}$ value. In the smoke layer, the
extinction Ångström exponent $A_\alpha$, can exceed $A_\beta$, due to negative contribution of $A_\beta^d$. In
particular, on 15 December $A_\alpha$ is about 1.1, while $A_\beta$ is close to zero.

To estimate the dependence of smoke lidar ratios $S_{355}$ and $S_{532}$ on RH, five smoke episodes

on 14-15, 15-16, 22-23, 24-25 December 2015 and 19-20 January 2016  were analyzed. $S_{532}$ and
$S_{355}$, together with relative humidity and the $\dfrac{\beta_{532}^s}{\beta_{532}}$ ratio are summarized, for these episodes, in
Table 1. The heights chosen correspond to the values of relative humidity close to maximum. The
calculated values of RH are characterized by high uncertainties, because lidar and sonde
measurements are not collocated. Estimations oof corresponding uncertainties are also given by
Table 1. The lidar ratio values from Table 1 are plotted in Fig.18 as a function of RH. These plots,
however, should be taken with care, because, for different days the smoke particles could have
different chemical composition, thus results may depend not only on RH. Moreover, the dust
particles occurring in the elevated layers, as discussed, can introduce an additional ambiguity in
the results. On 15 December (04:00 – 06:00 UTC) the lidar ratio $S_{532}$=44±5 sr is quite low and
"drops out" of other sessions. Nevertheless, Fig.18 demonstrates a clear increasing trend of S with
RH, at both wavelengths. From this figure, one can also conclude that $S_{355}$ always exceeds $S_{532}$
and, that $S_{532}$ for smoke can be as small as 44±5 sr at low humidity. The small values of $S_{532}$ for
the "fresh smoke" (about 40 sr) were reported also by (Burton et al., 2012).

To compare our observations with the lidar ratios used in the MERRA-2 model, we have

also performed the simulation of $S_{532}^{OC}(RH)$ and $S_{355}^{OC}(RH)$ dependence for organic carbon (OC)
based on the particle parameters and hygroscopic growth factor from MERRA-2 model. In
MERRA-2 the organic carbon is the main component of the biomass burning products. The
imaginary part of the OC increases in UV due to the presence of "brown carbon" (BrC), which is
a subset of organic carbon with strong absorption in the UV region (Bergstrom et al., 2007; Torres
et al., 2007). The majority of BrC is emitted into the atmosphere through low-temperature,
incomplete combustion of biomass. In the newest development of GEOS, biomass burning OC is





now emitted as a new BrC tracer species that uses $Im_{532}=0.009$ and $Im_{355}=0.048$ values (Hammer
et al. 2016). Thus, the spectral behavior of the imaginary part of organic carbon refractive index
depends on contribution of the BrC fraction to the primary organic carbon and on the physical-
chemical processes in the smoke layer during its transportation. As a result, the spectral
dependence of Im can present strong variations. In our study, the computations at 355 nm were
performed for four values of the imaginary part of dry particles $Im_{355}=0.048, 0.03, 0.02, 0.01$. At
532 nm two values $Im_{532}=0.005$ and 0.009 were considered. The parameters of the dry particle size
distribution, the real part of CRI and the hygroscopic growth factor used in computations are given
in Veselovskii et al. (2018). The particles are assumed to be homogeneous spheres and an increase
of the volume for every RH value (calculated from the growth factor) occurs due to water uptake.
Thus both the real and the imaginary part of CRI depend on RH.
The results of the simulations, shown in Fig.18, demonstrate strong dependence of the
organic carbon lidar ratio on the imaginary part of dry particles and on the relative humidity. For
$Im_{355}=0.048$, for all RH, $S_{355}$ is above 95 sr, which strongly exceeds the observed values. For lower
$Im_{355}$ the $S_{355}$ (RH) dependence is more pronounced and for $Im_{355}$ within the range 0.01-0.02,
computed $S_{355}$ are close to observed values. Computed $S_{532}$ values at low RH exceed the measured
ones, but for RH>70% agreement between measurements and GEOS assumed optical properties
for OC becomes reasonable.
The ratio $S_{355}/S_{532}$ for organic carbon, the same as for dust, is strongly influenced by the
spectral dependence of the imaginary part of CRI, hence it can be used as an indicator of Im
enhancement in UV. The ratios $S_{355}/S_{532}$, calculated from the results of modeling in Fig.18 for four
values of $Im_{355}$ (0.048, 0.03. 0.02, 0.01) and $Im_{532}= 0.009$, are shown in Fig.19. The ratio $S_{355}/S_{532}$,
corresponding to $Im_{355}=0.01$, is about 1.1 in the whole range of RH. However for the imaginary
part of dry particles $Im_{355}=0.02$ and 0.03 the ratio $S_{355}/S_{532}$ increases up to approximately 1.2 and
1.3 respectively for RH in 40%-70% range. Thus, enhanced $Im_{355}$ of dry OC particles should
provide increase of $S_{355}/S_{532}$ ratio even high RH and this is how it can be revealed. The measured
values of $S_{355}/S_{532}$ are shown on the same figure. As mentioned, observation at 532 nm on 15
December (RH=42%) "drops out" of other sessions. For the rest of observations the ratio $S_{355}/S_{532}$
is in 1.2 – 1.3 range, and from modeling in Fig.19 the imaginary part of the dry particles at 355
nm is estimated to be in 0.02-0.03 range.

**5. Summary and conclusion**
Our study shows the impact of aerosol spectral absorption variation on the lidar-derived
aerosol properties. In contrast to extinction, the backscattering coefficient, and so the lidar ratio,
are sensitive to the imaginary part of CRI. Hence, $S_{355}/S_{532}$ ratio can be an indicator of the





imaginary refractive index enhancement in the UV. Measurements performed during the
SHADOW campaign, in dust conditions, show a correlation between the decrease of $Im_{440}$, derived
from AERONET observations, and the decrease of lidar-derived $S_{355}/S_{532}$ ratio. Namely, in the
second half of April 2015, $S_{355}/S_{532}$ decreased from 1.5 to 1.0, when $Im_{440}$ decreased from 0.0045
to 0.0025. Our numerical simulations confirm, that observed $S_{355}/S_{532}$ (ratio close to 1.5) and $A_\beta$
(value close to -0.75) can be due to spectral variation of the imaginary part, attributed to iron oxides
contained in dust particles. Thus, April 2015 observations suggest the presence of different dust
types, characterized by distinct spectral dependence of $Im(\lambda)$. The analysis of backward
trajectories and Infrared Difference Dust Index derived from MSG geostationary satellite confirms
different air mass and dust particles transport features in the beginning and at the end of April.
Hence, the observed variations of $S_{355}/S_{532}$ can be related to the source region mineralogy. During
the April, particle depolarization systematically exceeded 30%, therefore no discrimination
between different types of dust was possible.

The results presented in this study demonstrate also that, for the selected temporal interval,
the dust lidar ratios may present significant variation with height. Dust of different size and
mineralogical composition can have different deposition rate, hence, complex refractive index can
be height-dependent. For instance, on April 1$^{st}$, the $S_{532}$ decreased with height from 60 sr to 50 sr
within 1000–3000 m range, while depolarization ratio exceeded 30%. The analysis of this episode
showed that variation of the lidar ratio is entirely attributed to variations of dust characteristics;
the smoke aerosol contribution was insignificant. The data also demonstrate that a seemingly
uniform dust layer may have quite a complex height variation. The results therefore suggest the
relevance of including a varying mineralogy in radiative and climatic modeling of desert dust
impacts.

During December – January, the dry season in western Africa, our observations allowed in
addition the analysis of biomass burning aerosol properties. These particles are a product of the
seasonal forest fires and intensive agricultural waste combustion and can contain a substantial
amount of organic compounds, characterized by an enhanced imaginary part in UV (so called
BrC). For this aerosol type, the $Im(\lambda)$ dependence should increase the lidar ratio at 355 nm and
influence $S_{355}/S_{532}$. The smoke particles can be also hydrophilic and the lidar ratio can therefore
exhibit a strong dependence on RH. The numerical simulations performed for organic carbon,
which is the main component of smoke in GEOS model, demonstrated that $S_{355}/S_{532}$ is close to 1.0
in the absence of spectral variation of the imaginary part; this ratio, however, can be as high as 1.8
for dry particles with $Im_{532}=0.009$ and $Im_{355}=0.048$. This $S_{355}/S_{532}$ ratio decreases with RH,
however even for high humidity it depends on the $Im_{355}$ value for dry particles. In particular, for
$Im_{355}=0.02$ and 0.03 the ratio $S_{355}/S_{532}$ is about 1.2 and 1.3 respectively for RH in 40%-70% range.





Thus, observed $S_{355}/S_{532}$ values, exceeding 1.0, could corroborate the enhancement of imaginary
refractive index for smoke in UV.

Several strong smoke episodes were observed during the SHADOW campaign. While we

were able to evaluate the RH profiles, the dependence of the smoke lidar ratio with RH has been
estimated. The results obtained should be taken as semi-qualitative only, due to possible variation
of smoke particles composition from episode to episode and due to the presence of dust particles.
Nevertheless, the results clearly demonstrate an increase of $S_{532}$ from 44±5 sr to 66±7 sr and of
$S_{355}$ from 62±6 sr to 80±8 sr, when the RH increased from 25% to 85%. The measured $S_{355}/S_{532}$
ratio varied mainly within the range 1.2 - 1.3, so comparison with modeling for OC provides the
estimate of $Im_{355}$ of dry smoke particles in 0.02-0.03 range.

We would like to conclude that the multi-wavelengths Raman and depolarization lidar

measurements in western Africa enabled quite unique and comprehensive profiling of dust and
smoke spectral absorption properties. The results demonstrated a high variability of the lidar ratio
and the presence of its spectral dependence. Our study is one of the first attempts to track aerosol
composition variability using lidar measurements and to understand the mechanism underlying the
observed variations.

**Acknowledgments**: The authors are very grateful to **t**he CaPPA project (Chemical and Physical
Properties of the Atmosphere), funded by the French National Research Agency (ANR) through
the PIA (Programme d'Investissement d'Avenir) under contract "ANR-11-LABX-0005-01" and
by the Regional Council" Nord-Pas de Calais » and the "European Funds for Regional Economic
Development (FEDER). We would like to acknowledge the AERONET team at NASA/Goddard
Space Flight Center in Greenbelt, MD, and Service National d'Observation PHOTONS from
University of Lille/CNRS/INSU operating under ACTRIS-FR research infrastructure, for
providing high-quality data. Development of lidar data analysis algorithms was supported by
Russian Science Foundation (project 16-17-10241).






Table 1. Lidar ratios S$_{355}$, S$_{532}$ for five smoke episodes in December 2015 – January 2016 and
corresponding the relative humidity RH. The table provides also the height and temporal interval
of observations. The contribution of the smoke particles to the total backscattering $\frac{\beta_{532}^{s}}{\beta_{532}}$ is derived
from depolarization measurements.

| Date | Height, m | Time, UTC | $\frac{\beta_{532}^{s}}{\beta_{532}}$ | RH, % | $S_{355}$, sr | $S_{532}$, sr |
|---|---|---|---|---|---|---|
| 15 Dec | 2000 | 04:00-06:00 | 0.57 | 42±8 | 67±7 | 44±5 |
| 15 Dec | 1850 | 19:20-20:30 | 0.57 | 25±6 | 62±6 | 50±5 |
| 23 Dec | 2250 | 05:00-07:00 | 0.65 | 65±13 | 76±8 | 56±6 |
| 24 Dec | 3200 | 19:00-23:00 | 0.66 | 75±14 | 76±8 | 62±6 |
| 20 Jan | 4500 | 01:00-07:00 | 0.8 | 85±15 | 80±8 | 66±7 |






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



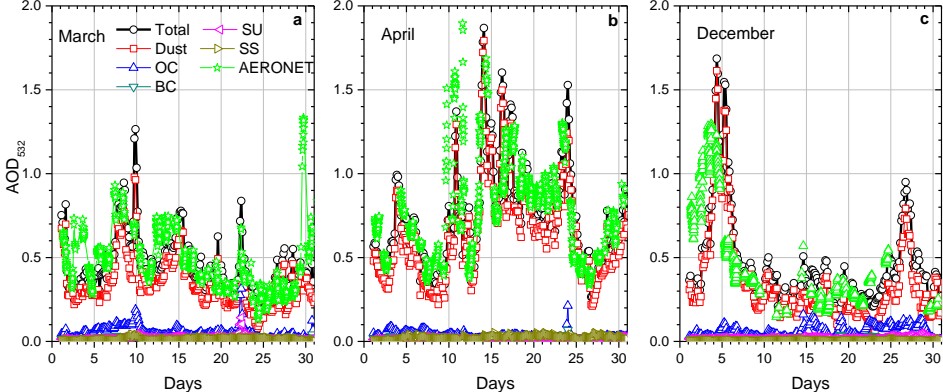

Fig.1. The aerosol optical depth (AOD) at 532 nm (open circles) and AODs of the main aerosol components, such as dust, organic carbon (OC), black carbon (BC), sulfates (SU) and sea salt (SS) provided by the MERRA-2 for (a) March, (b) April and (c) December 2015 over Mbour. Open stars show $AOD_{532}$ provided by AERONET.

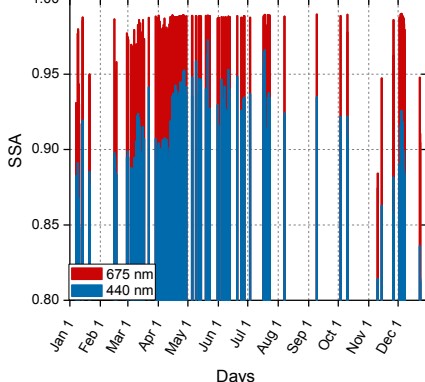

Fig.2. Aerosol single scattering albedo (SSA) at 675 nm and 440 nm provided by AERONET for M'bour site in 2015.






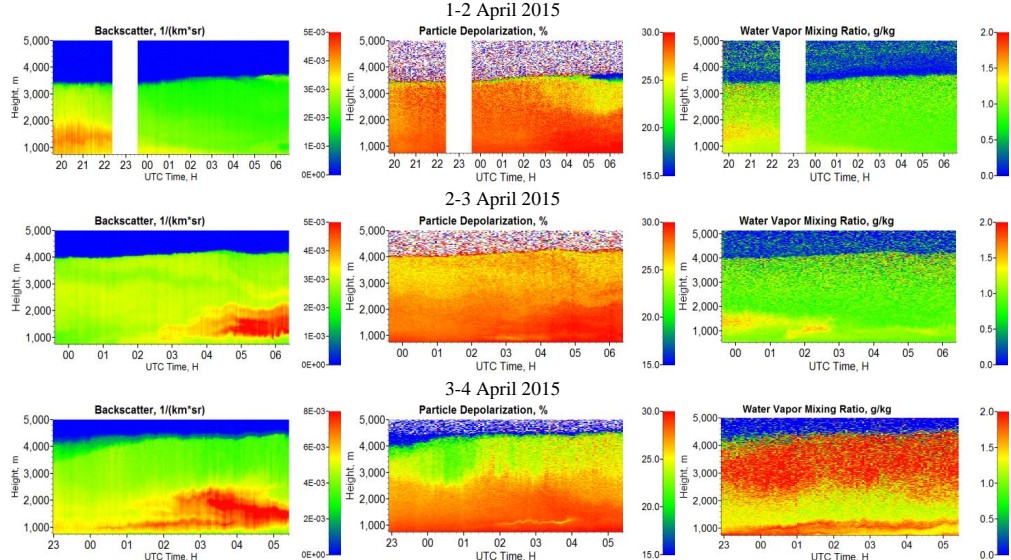

Fig.3. Tempo-spatial distributions of aerosol backscattering coefficient $\beta_{532}$ (left column), particle
depolarization ratio $\delta_{532}$ (middle column) and water vapor mixing ratio (right column) for the
nights 1-2 April (upper row), 2-3 April (middle row) and 3-4 April 2015 (bottom row).



02 April 03:00          03 April 03:00          04 April 03:00

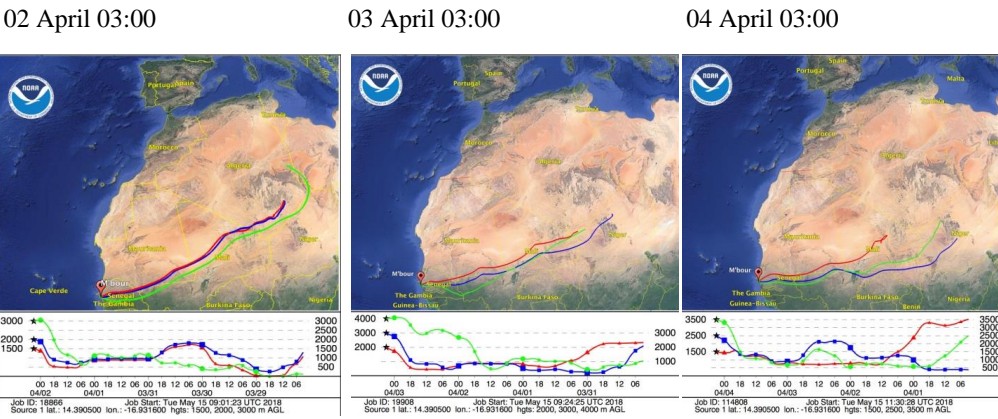

Fig.4. Three-day backward trajectories from the NOAA HYSPLIT model for the air mass in
M'bour on 2, 3, 4 April 2015 at 03:00 UTC.

781





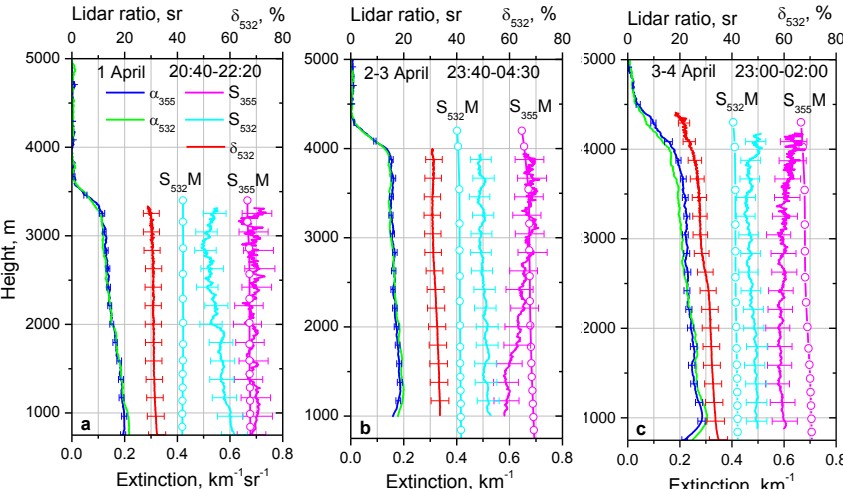

Fig.5. Vertical profiles of extinction coefficients ($\alpha_{355}$, $\alpha_{532}$) and lidar ratios ($S_{355}$, $S_{532}$) at 355 nm and 532 nm together with particle depolarization ratio $\delta_{532}$ measured on 1 April (20:40-22:20 UTC), 2-3 April (23:40-04:30 UTC) and 3-4 April 2015 (23:00-02:00 UTC). Symbols show the lidar ratios of dust provided by MERRA-2 model ($S_{355}M$, $S_{532}M$).

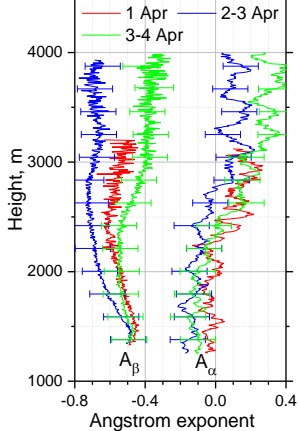

Fig.6. Vertical profiles of the extinction and backscattering Ångström exponents ($A_\alpha$ and $A_\beta$) at 355 – 532 nm for three temporal intervals from Fig.5.




792

793

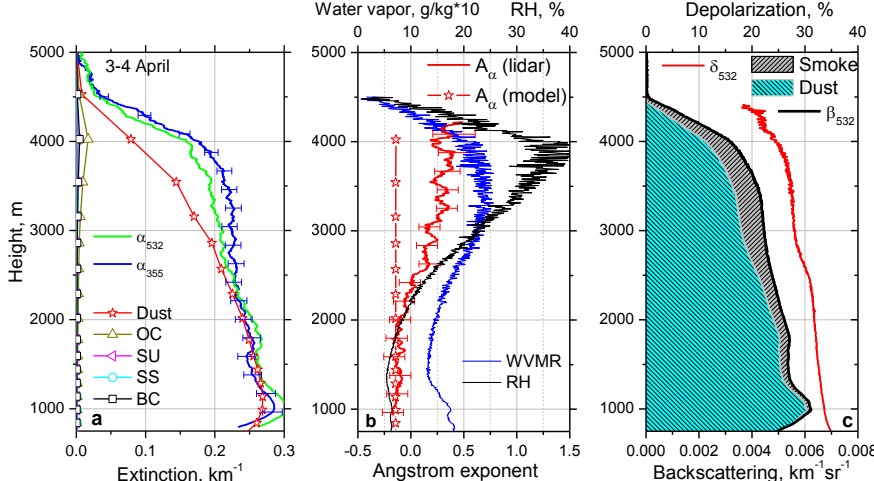

Fig.7. Vertical profiles of (a) extinction coefficients at 355 nm and 532 nm ($\alpha_{355}$, $\alpha_{532}$) measured by lidar (lines) and modeled by MERRA-2 (line+symbol) for five aerosol components at 532 nm; (b) extinction Ångström exponents at 355-532 nm obtained from lidar observations and modeled by MERRA-2 for pure dust (stars) together with water vapor mixing ratio (WVMR) and the relative humidity; (c) contribution of dust and smoke particles to $\beta_{532}$ together with particle depolarization ratio $\delta_{532}$. Values of WVMR are multiplied by factor 10. Lidar measurements were performed on 3-4 April 2015 for period 23:00 – 02:00 UTC. Modeling results are given for 4 April 00:00 UTC.










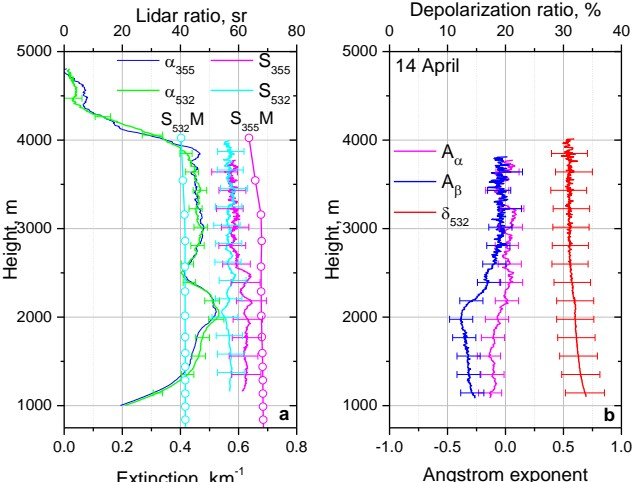

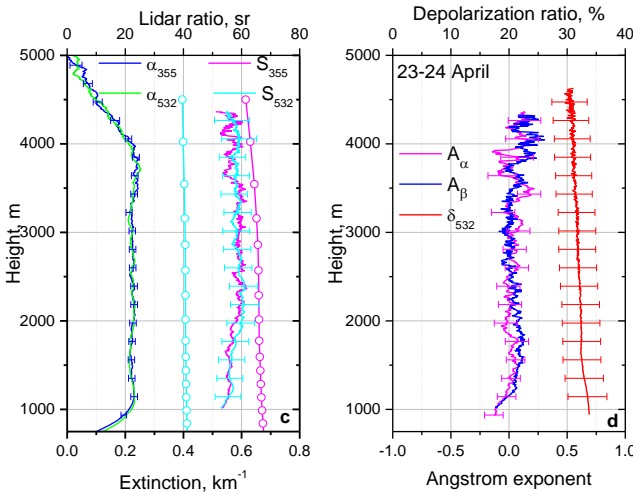

Fig.8. (a, c) Vertical profiles of extinction coefficients ($\alpha_{355}$, $\alpha_{532}$) and lidar ratios ($S_{355}$, $S_{532}$) at 355 nm and 532 nm; together with (b, d) particle depolarization ratio $\delta_{532}$, and extinction and backscattering Ångström exponents ($A_\alpha$, $A_\beta$) measured on (a, b) 14 April 2015 (00:00 – 05:00 UTC) and (c, d) the night 23-24 April (23:00-06:00 UTC). Open symbols on plots (a, c) show the lidar ratios $S_{355}M$ and $S_{532}M$ provided by MERRA-2 model on 14 and 14 April at 00:00 UTC.








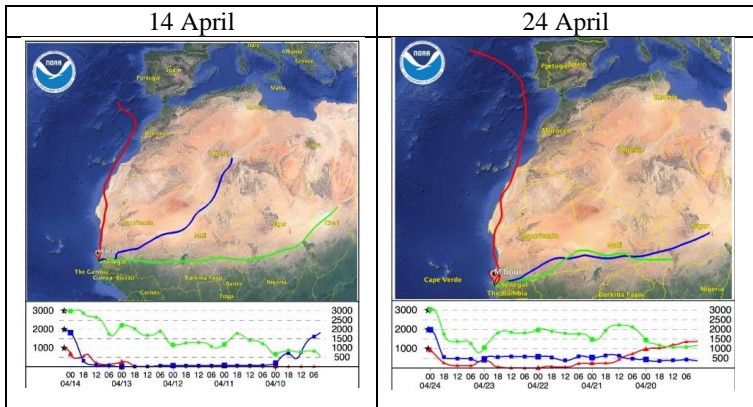


Fig.9. Four-days backward trajectories from the NOAA HYSPLIT model for 14 April (03:00 UTC) and 24 April (00:00 UTC) 2015.






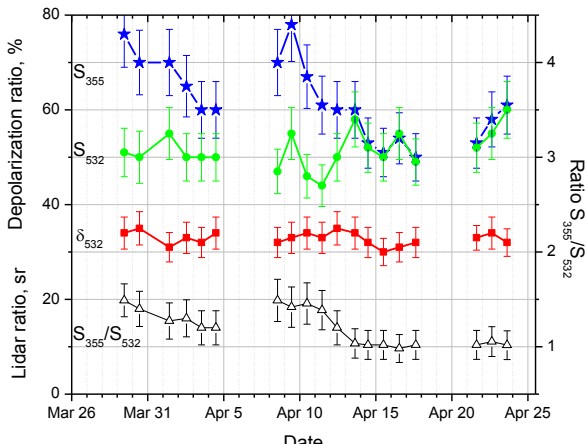

Fig.10. Lidar ratios $S_{355}$, $S_{532}$ and the particle depolarization ratio $\delta_{532}$ for dust episodes in March
- April 2015. Open triangles show the ratio $S_{355}/S_{532}$.

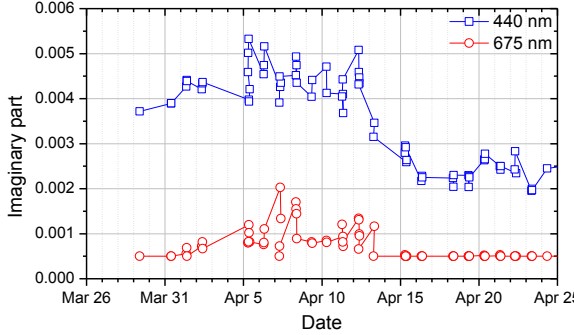

Fig.11. Imaginary part of the refractive index at 440 nm and 675 nm provided by AERONET in
March – April 2015






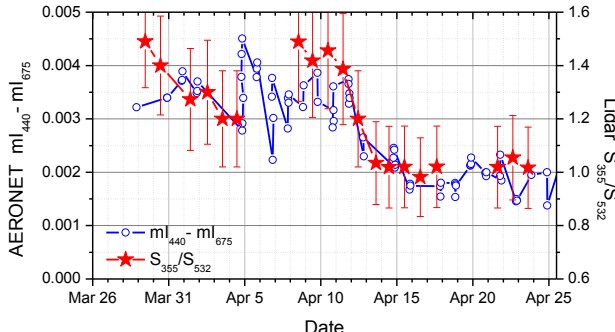

Fig.12. Difference $Im_{440}$ - $Im_{675}$ from Fig.11 together with lidar measured values $S_{355}/S_{532}$ from
Fig.10 for days in April 2015.

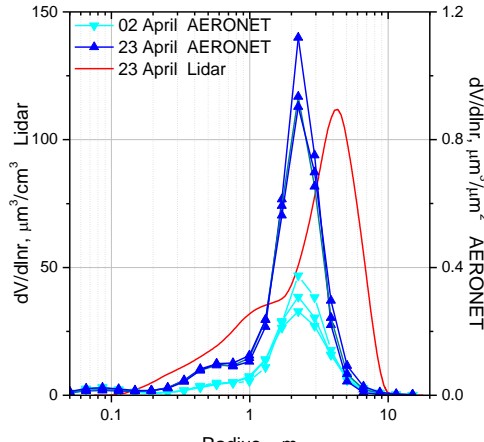

Fig.13. The particle size distributions provided by AERONET on 2 and 23 April 2015 (three PSDs
for each day). Red line shows the PSD derived from $3\beta+2\alpha$ lidar measurements on 23-24 April.






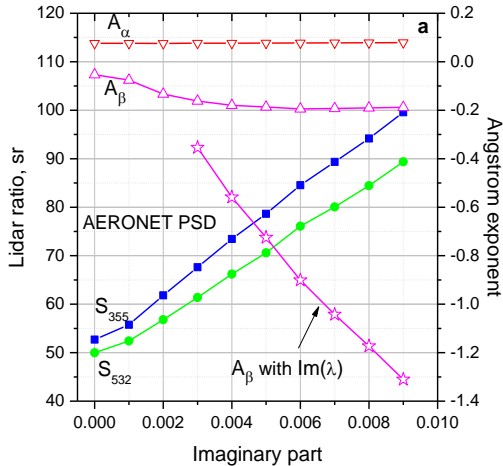


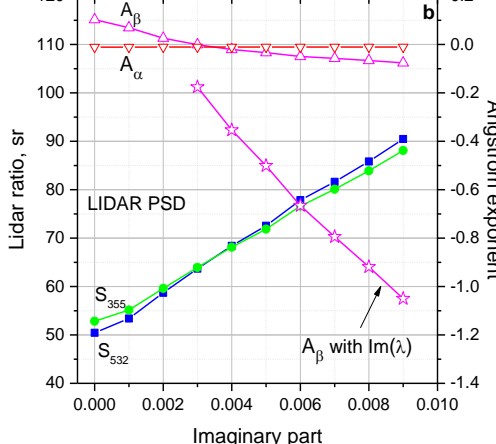

Fig.14. Lidar ratios $S_{355}$, $S_{532}$ together with the extinction and backscattering Ångström exponents
$A_\alpha$ and $A_\beta$ calculated for (a) AERONET PSD on 23 April from Fig.13 and (b) lidar derived PSD
from Fig.13 as a function of the imaginary part. Open stars show $A_\beta$ for spectrally dependent
imaginary part Im($\lambda$), assuming that Im$_{532}$=0.002 is fixed and only Im$_{355}$ is free vary. Computations
are performed for the assembly of randomly oriented spheroids with the real part Re=1.55.



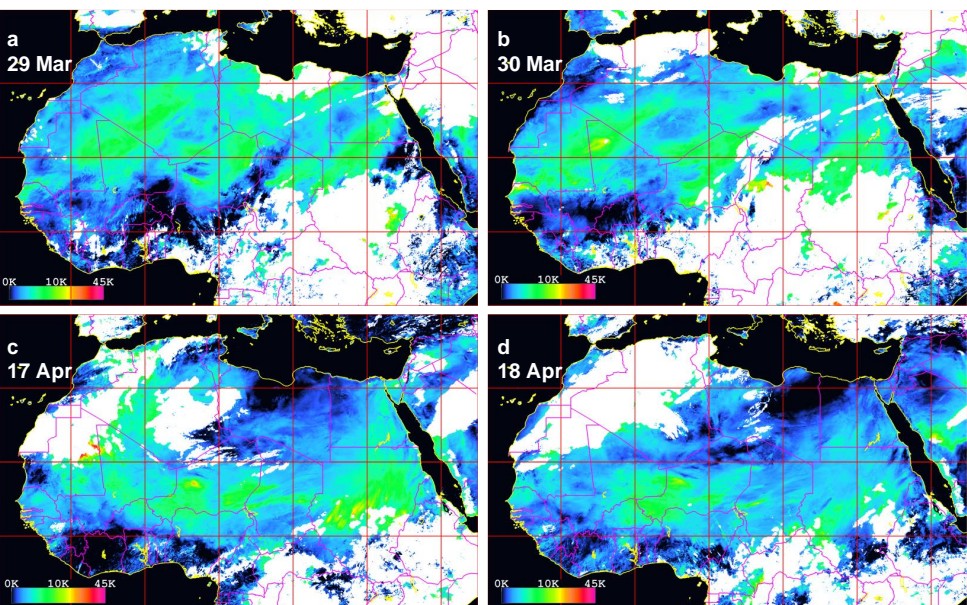

Fig.15. Infrared Difference Dust Index (IDDI) derived from MSG geostationary satellite at noon time. Panels (a), (b) show IDDI elevated values, representing airborne dust emission and transport, over central and northern Sahara on 29, 30 March 2015. The dust transport regime is visibly changed a few days later (17, 18 April 2015, panels (c), (d)); the elevated IDDI values are shifted to the south. The areas in white are cloud screened pixels; the IDDI is derived only over land due to the algorithm physical principle.






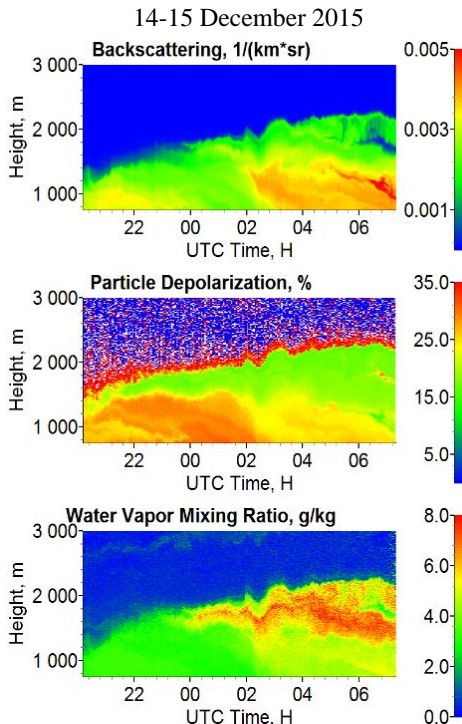

Fig.16. Tempo-spatial distributions of aerosol backscattering coefficient $\beta_{532}$, particle
depolarization ratio $\delta_{532}$ and water vapor mixing ratio during smoke episode on the night 14-15
December 2015.






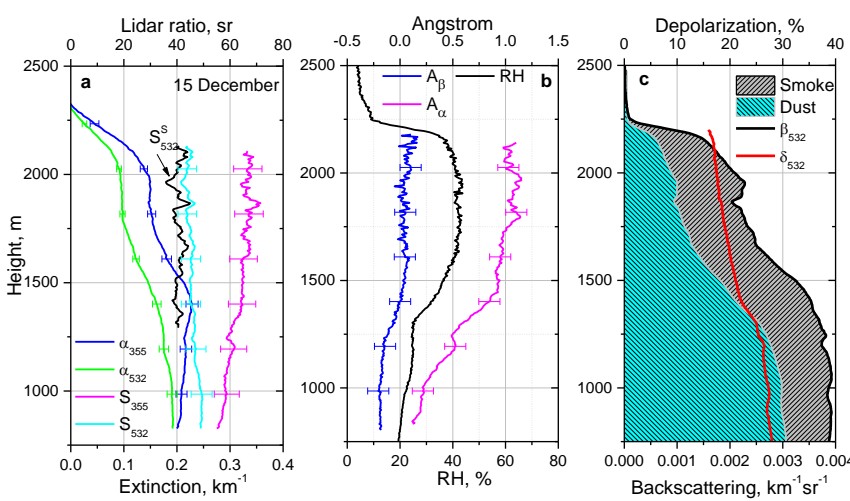

Fig.17. Vertical profiles of (a) extinction coefficients ($\alpha_{355}$, $\alpha_{532}$) and lidar ratios (S$_{355}$, S$_{532}$); (b)
extinction, backscattering Ångström exponents (A$_\alpha$, A$_\beta$) at 355 − 532 nm and relative humidity
RH; (c) contribution of dust and smoke to $\beta_{532}$ together with particle depolarization ratio $\delta_{532}$ on
15 December (04:00 − 06:00 UTC). Black line in plot (a) shows the lidar ratio of smoke $S_{532}^{s}$
calculated from Eq.5.






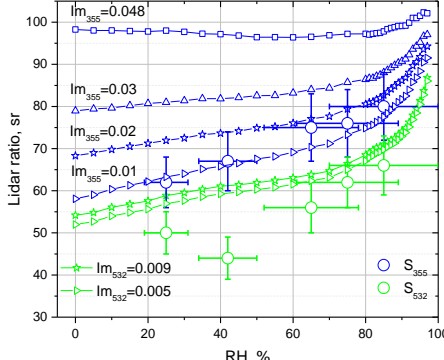

Fig.18. Modeled lidar ratios of organic carbon at 355 nm and 532 nm (line + symbol) as a function
of the relative humidity for the particle parameters used in the MERRA-2 model. At 355 nm results
are given for four values of the imaginary part of dry particles: $Im_{355}$= 0.048, 0.03. 0.02, 0.01. At
532 nm two values $Im_{532}$= 0.009 and 0.005 are considered. The scattered symbols (circles) show
the lidar ratios ($S_{355}$, $S_{532}$) observed during five smoke episodes from Table 1.

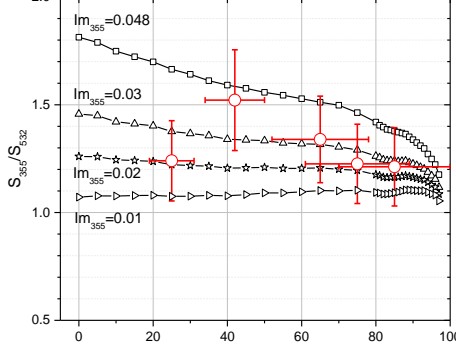

Fig.19. The ratio $S_{355}/S_{532}$ for organic carbon as a function of the relative humidity calculated from
modeling results in Fig.18 for $Im_{532}$=0.009 and $Im_{355}$= 0.048, 0.03. 0.02, 0.01. The scattered
symbols (circles) show the observed $S_{355}/S_{532}$ values.