# Peer review of "Variability of Lidar-Derived Particle Properties Over West Africa Due to Changes in"

_Atmospheric Chemistry and Physics, 2020_

## Referee Comment (RC1) · Anonymous Referee #3 · 18 Mar 2020

The paper "Variability of Lidar-Derived Particle Properties Over West Africa Due to Changes in Absorption: Towards an Understanding" presents and discusses the possibility to identify the spectral dependence of dust refractive index imaginary part (form here on iDRI) using Raman lidar measurements. Goal of this analysis is to reveal the effect of dust increased absorption in the UV on lidar derived parameters. The authors use 3+2+1 lidar measurements performed during the "SHADOW-2" campaign in Senegal, as well as the available AERONET dataset for the campaign period. The analysis is performed for dust dominated cases during April 2015 and it is separated between two periods: first and second half of April, based on variations of SSA440 derived from AERONET. More specifically, during the second period SSA440 increases indicating

that dust particles become less absorbing.

The analysis is not only limited to dust dominated cases but as supplementary subject smoke lidar ratio variability is examined in relation to relative humidity variations. Furthermore, the smoke S355/S532 ratio is examined in order to provide indications of increased smoke absorption in the UV. To this end case studies during December 2015 and January 2016 are selected since during this period intense forest fires are observed in the region.

In general, I find this study very interesting and of high value and I believe it falls within the scope of ACP. The authors have done a thorough job in presenting the results, the manuscript is well-written / structured, the presentation clear and the quality of the figures high. Furthermore, the authors give credit to related work and the results support the conclusions. However, in order to help improving the manuscript, I would kindly suggest the authors to take into account the following minor comments:

Fig.8a and Page 9, Lines 283-285: "Thus, we can assume that increase of the imaginary part in UV in the first layer is more significant, than in the second one".

I noticed from this figure that the Angstrom exponent (both backscatter and extinction related) increases towards higher altitudes, which coincides with a slight decrease of depolarization ratio and a coincidence of the S355 and S532. Could these variations point towards the dominance of smaller dust particles higher in the layer? From laboratory studies we know that smaller dust particles present lower depolarization ratio values (i.e. Järvinen et al. 2016; Sakai et al., 2010), while also the larger S532 values lower in the layer could be attributed to the increased "sensitivity" this wavelength should have to the presence of larger particles. Why is the dominance of smaller dust particles should be excluded here?

Page 8, Line 245: "assuming 35% and 7% for dust and smoke depolarization ratio".

Please provide some references on choosing these values specifically. Did you use

the same values also for the dust-smoke decomposition you perform for all the cases presented?

Also Page 14, Line 429: "In principle, we can estimate $S\_532\hat{}s$ using Eq.5, because the ratio $(\beta\_532\hat{}s)/\beta\_532$ is available". Here the $\beta\_532\hat{}s$ depends on the selected value of $d\_532\hat{}s$. Can you provide an estimation of the uncertainties of this approach? What could be the effect on the resulting $S\_532\hat{}s$ values?

Table 1: Could you add to the table the height intervals chosen for the analysis of the smoke layers and also an estimation of their lifetime? Could any differences in the smoke properties be related to the age of smoke particles?

Page 4, Line 113: for the range resolution of particle extinction coefficient it is not clear to me which height intervals are selected. Do you mean 50 m up to 1000m and 125 m from 1000m to 7000m?

Page 7, Line 217: the authors probably mean "hydrophobic".

Page 11, Line 344: "spectrally independent refractive index". Please provide the selected values for this analysis.

Page 12, Line 369: "so variation of the imaginary" add "part of the refractive index".

Section 3.2: Please provide the spatial-temporal evolution of backscatter coefficient, water vapor and particle depolarization for these cases also.

Please also note the supplement to this comment:
https://www.atmos-chem-phys-discuss.net/acp-2020-98/acp-2020-98-RC1-supplement.pdf

---

## Referee Comment (RC2) · Anonymous Referee #4 · 31 Mar 2020

This work deals with the analysis of spectral variation on lidar ratios and their relationships with spectrally dependent refractive index. The topic is very interesting in the field of atmospheric sciences because will help to further advances in constraining aerosol absorption properties from lidar measurements alone. The focus is on Saharan dust particles and which is one of the largest sources in the world of natural aerosol. Authors related quite well differences in spectrall dependences of complex refractive index with possible variability of dust composition. Actually, these results contribute to further advance in dust properties as in many previous studies with remote sensing dust have been assumed as an homogeneous specie independently of the source. The present papers also analysed mixtures of dust with biomass burning and how lidar ratios vary

in such mixtures.

The authors use an unique datasets for SHADOW-2 field campaings developed in Africa, and uses a high-quality data from LILLAS multiwavelength lidar system. The analysis presented complements previous studies carried out by the authors in Africa Currently, aerosol properties measurements in the African continent are very sparse and SHADOW-2 measurements are contributint undoubtlly to fill that gaps.

MINOR REVISIONS

I have a personal curiosity related to the paper: The authors have a lot of works on retriving aerosol microphysical properties from 3b+2a lidar measurements, even for non-spherical particles. Here, authors have the measurements for the retrieval. I would like to know why authors have decided not to do the 3b+2a inversion to retrieve aerosol refractive index. I have also follow your last papers in retrievals of aerosol microphysical properties from space-borne simulations and I would like to know if your new results can have an impact in space retrievals.

In the discussion of changes in lidar ratios for smoke with relative humidity, please take into account that they do not only depend on refractive index. Also is important the possible changes in size distribution.

Line 51-55: Please, note that spectral dependence in lidar ratio have been demonstrate useful to estimate the range of refractive index for non-spherical particles, although assuming no spectral dependence in CRI

Line 77: Please, define variables. What is Im440 and Im355.

Line 110: Why measurements are acquired at 47 degrees angle to horizon?

Equation 1: Please, give a proper reference

Line 163: I thing there is a type in 'recalculated'

Line 204: ' The backcatter Ångström exponent Ab, in contrast with Az is sensitive

to the spectral dependence of the imaginary part of CRI'. Please clarify and provide references. Actually, Aa also depends on imaginary refractive index.

Line 245: Please explain futher why you assume 35% and 7% for dust and depolarization ratio or provide appropiate references.

Line 258: It is not clear why 25 sr is unrailstic lidar ratio. Please, provide references.

Line 267: Why smoke lidar ration should increase with RH ?

Line 276: the statement 'dust became less absorbing in the UV' is unclear. Are you referring to imaginary refractive index or to single scattering albedo?

Lines 338-353: Authors give a description of the 3b+3a lidar inversion. That should come earlier becase previously in Figure 13 you show size distribution from 3b+2a inversion.

Lines 333-337: It is not clear to me how you make the simulations. Table 1: How you estimated uncertainties?

Lines 465-467: Please, note the limitations in PSD variability with relative humidity in MERRA-2.

Conclusion section: 'Our study shows the impact of aerosol spectral absorption variation on the lidar-derived aerosol properties'. Do you refer to any aerosol type? I think you want to say dust and smoke aerosol.

Figure 7a. There is a blue line missing.

---

## Referee Comment (RC3) · Anonymous Referee #1 · 1 Apr 2020

[referee-annotated manuscript omitted]

---

## Referee Comment (RC4) · Anonymous Referee #2 · 15 Apr 2020

The manuscript contains a well-elaborated scientific study about the link between lidar-derived optical properties of dust and dust-smoke mixtures in western Africa and the basic aerosol microphysical and dust chemical composition. The manuscript is well written and appropriate for ACP. The key part deals with the relationship between measured extinction-to-backscatter ratios at two wavelengths (and corresponding wavelength dependence) and the imaginary part of the refractive index. However, conclusions are drawn (from observations and simulations) which are not at all trustworthy (to this reviewer) as long as a spheroidal shape model for the unknown irregular shape of mineral dust particles must be assumed in the modelling approaches and the influence of this assumption on the final results remains unknown, at least is not discussed and

quantified. Another critical point is that a reader may get the impression that the findings hold for mineral dust in general, and not only and specifically for western Saharan dust and dust/smoke mixtures. However, the literature contains numerous papers that are in contradiction with the findings presented here and in Veselovskii et al. (2016). For example, such large values of up to 1.5 for the ratio of the 355 to 532nmlidar ratios as reported here together with particle depolarization ratio of larger than 30 % and thus indicating pure dust conditions has never been reported in numerous others papers in the literature. Larger backscatter in the green than in the blue is reported sometimes but only as an exception. Furthermore, all these inversion procedures(lidar, photometer) assume particles with radius < 15 $\mu$m, but the lidar field site (Senegal) is almost in a desert source region so that the probability of the occurrence of giant particles is never zero. This can be another 'efficient' error source.

Detailed comments and suggestions:

P2 L65-67: One should even include the third part of the entire SAMUM-SALTRACE series and the related papers of Gross et al 2015, Rittmeister et al., 2017, and Haarig et al. 2017.

P2 L69-77: I am confused that the backscatter coefficient (a pure scattering parameter) is very sensitive to the imaginary part of the refractive index (describing the absorption efficieny...), but the extinction coefficient (absorption plus scattering coefficient) is insensitive to the imaginary part (to say it again.... describing the absorption features of aerosol particles). Can you please provide more background information on this 'apparent contradiction' that a scattering process is influenced by the imaginary part, but absorption features not?

P2 L126: The paper emphasis the usefulness of the lidar ratio (S) wavelength dependence. Why not writing down the relationship between all the different Angstrom exponents, AE(ext), AE(bsc), AE(S), that is AE(S) = AE(ext) – AE(bsc) as introduced by Ansmann et al., JGR, 107, 10.1029/2001JD001109, 2002. In this way, it is much

more easy to follow the discussion later on, when AE(ext) is frequently around zero, because then we have simply: AE(S) = -AE(bsc).

P4 L120 – P5 L138, A lot of equations. . ... Are all these quantities used later on?

P7 L206 – 209: Here again the confusing relationship between backscatter coefficient and imaginary part . . ... I am still confused by the high negative AE(bsc) values. Unfortunately, extinction profiles (with rather reasonable wavelength dependence) are frequently shown, but the profiles of the backscatter coefficients for these high negative AE values are never shown. Such observations are rare in the literature and may be related to the region of western Africa and proximity to dust source regions (and the omnipresence of coarse and giant dust particles).

The authors argue, it has to do with the chemical composition of the dust particles, and then with the imaginary part? But my question is continuously: If we use an alternative shape model what would then be the result. . .? Gasteiger et al., and modelling groups in Finland introduced very different shape models. . ... And found very different results in terms of dust optical properties. And if we ignore all the giant dust particles in the complex data analysis, how large can be the damage?

P7 l 210-220: The lidar ratios for dust (355 vs 532 nm) are so different (60-70sr vs 45-60sr), although the depolarization ratio shows (almost) pure dust conditions, this is a very surprizing result and in contradiction with the literature (Tesche et al, 2011, Gross et al., 2011). Any explanation for this? Maybe. . . it has to do again with missing coarse and giant, irregularly shaped dust particles? Because AE(S) = - AE(bsc), why is backscatter Angstroem exponent so pronounced, why is that so negative? Is everything ok with the backscatter retrieval (calibration in the reference height)? My explanation would be: smoke is responsible for the strong lidar ratio wavelength dependence, but then the depolarization ratio should indicate that.

P9 L266-267: There are also smoke lidar ratios from the SAMUM Cabo Verde campaign (Tesche, Tellus, 2011, second paper).

P11 L325-326: The lidar ratio depends on complex refractive index and size distribution, BUT ALSO ON SHAPE of the particles. Again the conclusion from my side: Are all the simulations and retrieval products trustworthy and reflect the reality when a spheroidal shape model is needed and used? Can we trust the main findings and conclusions? And again: What about the impact of missing giant, irregularly shaped particles? They are there over-near source regions!

P11 L355-356: observed low values of AE(bsc) cannot be produced without accounting for a spectral dependence of the imaginary part. . ... Yes, for the case of the assumed spheroidal shape model and size distribution up to about 10$\mu$m radius.

AERONET is used for comparison. But here (photometer) also a spheroidal shape model is needed for dust particles. So, no independence between lidar and photometer products.

P12 L368-369: . . . the authors write. . .again. . .: . . .. the observed S355/S532 ratio and AE(bsc) can be explained by the spectral dependence of the imaginary part of CRI. My answer again: Yes, for spheroidal particles. So again. . . the reader is left with the question, and what is now the true impact of the imaginary part. . ... for irregularly shaped particles for which we do not have a model?

P12 L371: S355/S532 is 1.5 !!! and the depol ratio >0.3 indicates pure dust! Such a result I have never seen in the literature, and thus must have to do with the field site (and probably to the omnipresence of very large dust particles). I have no other explanation. Disregarding, whether my comment makes sense or not, the findings in this paperseem to be specific for Senegal, or maybe regions in the vicinity of dust source regions. That should be clearly written.

P13 L418: The authors write: The relative humidity varied from episode to episode. . ... To my knowledge, different RH conditions usually show different air masses and different aerosol properties from different sources. At least it seems to be a difficult task to correlate RH with found aerosol properties, and to draw trustworthy conclusions on

the hygroscopic (water uptake) properties.

P14, L447-448 Ok, this is said now but that comes very late.

P14 : When modelling lidar ratios for smoke, do you have proper smoke particle size distributions as input?

P15 L476-478: Also the size distribution changes a bit when the relative humidity increases. Is that considered in the simulations?

P17 after L545-550: At, at the end of the summary and conclusion section, one could mentioned as an outlook, that such studies as presented here should be repeated at many different places around the world, e.g. in the Middle East, Central and East Asia, Australia, and North Amerika. . ... in order to improve our knowledge on real-world aerosol optical properties needed in optical modelling and climate relevant atmospheric modelling.

My final summarizing comment: Disregarding the criticisms made here, my overall impression is: The discussion and sensitivity studies are nice and very helpful to interpret sophisticated multiple wavelength polarization Raman or HSRL lidar observations of complex dust and dust-smoke mixtures. The authors are experts and contribute with an important contribution to the field of dust atmospheric studies and interpretation in terms of dust size distribution, shape properties, and chemical composition. However, the discussion should be more sensitive to all the shortcomings (unknown shape, ignored giant particles). The more clearly the short comings are presented the more exciting the story and the probability to trigger further studies. The field is interesting, and the more questions are left at the end, the more interesting is this research field for the next generation of researches. To define the right questions is often better then to present some (questionable) answers. . .

Figures:

Fig.1: Please skip the symbols, just show different colored thin lines. It is hard to see

anything, except the ups and downs in the curves.

Fig. 5a: The lidar ratios of 70 sr at 355 nm and 50 sr at 532 nm clearly shows the strong impact of smoke to my opinion. The extinction profiles and the depolarization ratio > 0.3 indicate pure dust! I am very much confused.

Fig.5b: The same here.... And the modelling results are at all even more extreme ... and thus totally unrealistic to my opinion .... with the spheroidal model as input...almost 40 sr at 532 nm and 70 sr at 355 nm for pure dust conditions! Has this something to do with any dust reality? And AE(S) of 1.2 or and AE(Bsc) of -1.2 from the model?

Fig 6: Here one could discuss AE(S)=AE(ext)-AE(bsc) when showing AE(ext) and AE(bsc). AE(ext) atleast shows what we know from AERONET and the many dust lidar observations. But the backscatter wavelength dependence remains unique and seems to be quite special for this part of the world, whatever the reason may be.

Fig. 8: Again, nicely measured 355 and 532 nm extinction profiles (a), and (surprisingly) 'well-known' lidar ratios in the upper part of the dust layer, and very trustworthy depol ratios (b) and trustworthy AE(ext) profile (b), and even AE(bsc) and thus AE(S) in the upper part of the layer, the same for (c) and (d). This is what we know from the literature and all the SAMUM and SALTRACE findings for Saharan dust and aged outflow dust transported to the west.

And then again this huge contrast: AGAIN these strange model results. Lidar ratios up to almost 70sr at 355 nm and down to 40sr at 532 nm, and at the same time, these very reasonable lidar measurements. The authors seem to provide me (voluntary) with the wappons I need to blame the modelling effort and point on the used shape model as the main source for these errors. What shall we believe at the end, from all the conclusions the author derive, when we see this? Fig.8 is an excellent figure to corroborate my opinion: Ido not believe in any of the result!

Fig 10: The depolarization ratio shows no change in the dust conditions over the plotted one month period and is so high (PURE DUST), but the lidar ratio is sensitively changing. . .., and again probably solely dependent on the backscatter coefficient which is a strong function of refractive index and particle shape! So, what to do? Ok there is a correlation with the refractive index from AERONET. But AERONET also assumes a spheroidal model to produce dust-related results. So, what can we believe at the end.

Fig 12: Similar curves , good! But what does it help?

Fig 13: Size distribution from AERONET and from lidar (at what height?). Differences are visible but both size distributions stop at 10 microns. What about bigger particles when measuring very close to the source region. . . Is it possible that the used size distribution in all model runs underestimate the impact of the giant particles?

Fig. 14: I do not believe the Imag curve (stars) as long as we have another errors source (the spheroidal shape model). AE(Bsc) = - AE(S) can be used her in the discussion because AE(ext) is almost zero.

Fig 17-Fig 19: The RH study and discussion makes the paper quite long, and all the efforts are again quite speculative. . . Fig 19 is not convincing.

---

## Author Comment (AC1) · 22 Apr 2020

Response to reviewer 2

First of all, we would like to thank the reviewer for very detailed review and numerous useful comments. We tried to include these in revised manuscript.

Below are our responses:

*However, conclusions are drawn (from observations and simulations) which are not at all trustworthy (to this reviewer) as long as a spheroidal shape model for the unknown irregular shape of mineral dust particles must be assumed in the modelling approaches and the influence of this assumption on the final results remains unknown, at least is not discussed and quantified.*

**Discussions**

*Another critical point is that a reader may get the impression that the findings hold for mineral dust in general, and not only and specifically for western Saharan dust and dust/smoke mixtures. However, the literature contains numerous papers that are in contradiction with the findings presented here and in Veselovskii et al. (2016). For example, such large values of up to 1.5 for the ratio of the 355 to 532nm lidar ratios as reported here together with particle depolarization ratio of larger than 30 % and thus indicating pure dust conditions has never been reported in numerous others papers in the literature.*

Reviewer is right, there are numerous publications where researches reported, that lidar ratios at 355 and 532 nm coincide. We should mention, that recent measurements of Saharan dust at Crete also show LR355>LR532. Actually, the goal of this manuscript is to show that lidar ratios (LR) may coincide (LR355=LR532) or LR355>LR532 nm depending on dust origin. We added corresponding comment in Conclusion.

*Larger backscatter in the green than in the blue is reported sometimes but only as an exception.*

Yes, I don't think that these observations are done for pure dust.

*Furthermore, all these inversion procedures(lidar, photometer) assume particles with radius < 15 _m, but the lidar field site (Senegal) is almost in a desert source region so that the probability of the occurrence of giant particles is never zero. This can be another 'efficient' error source.*

In the measurements of the lidar ratio we don't make assumption about particle shape or size. But in AERONET, the giant particles may be provide some effect. Unfortunately, at a moment we are not able to quantify it.

**Detailed comments and suggestions**:

*P2 L65-67: One should even include the third part of the entire SAMUM-SALTRACE series and the related papers of Gross et al 2015, Rittmeister et al., 2017, and Haarig et al. 2017.*

Added

*P2 L69-77: I am confused that the backscatter coefficient (a pure scattering parameter) is very sensitive to the imaginary part of the refractive index (describing the absorption efficieny), but the extinction coefficient (absorption plus scattering coefficient) is insensitive to the imaginary part (to say it again describing the absorption features of aerosol particles). Can you please provide more background information on this 'apparent contradiction' that a scattering process is influenced by the imaginary part, but absorption features not?*

Yes, this is property of backscattering, and it does not depend on particle shape. This is true both for spheres and spheroids. Corresponding computations can be seen, for example in Fig.6. of (Veselovskii et al., 2010).

[Figure]

**Figure 6.** The particle backscattering (circles) and extinction (stars) coefficients at 355 nm wavelength as a function of the imaginary part of the refractive index. Calculations for spheres (open symbols) and spheroids (solid symbols) were performed for the $PSD_{20}$ with $m_R = 1.55$, $N_f = 100$ $cm^{-3}$.

The extinction is less sensitive, because it is the sum of scattering and absorption: with increase of the imaginary part the absorption rises, but scattering decreases. Reference is added to manuscript.

*P2 L126: The paper emphasis the usefulness of the lidar ratio (S) wavelength dependence. Why not writing down the relationship between all the different Angstrom exponents, AE(ext), AE(bsc), AE(S), that is AE(S) = AE(ext) – AE(bsc) as introduced by Ansmann et al., JGR, 107, 10.1029/2001JD001109, 2002. In this way, it is much more easy to follow the discussion later on, when AE(ext) is frequently around zero, because then we have simply: AE(S) = -AE(bsc).*

Yes, it is possible to introduce Angstrom exponent for lidar ratios also. Still we prefer to show ratio $S_{355}/S_{532}$, think it is more convenient.

*P4 L120 – P5 L138, A lot of equations: Are all these quantities used later on?*

Yes, when we analyze contribution of smoke, all these equations are used.

*P7 L206 – 209: Here again the confusing relationship between backscatter coefficient and imaginary part : I am still confused by the high negative AE(bsc) values. Unfortunately, extinction profiles (with rather reasonable wavelength dependence) are frequently shown, but the profiles of the backscatter coefficients for these high negative AE values are never shown. Such observations are rare in the literature and may be related to the region of western Africa and proximity to dust source regions (and the omnipresence of coarse and giant dust particles).*

We didn't show profiles of backscattering coefficients because extinctions and lidar ratios are presented. But in the revised manuscript we added Fig.6, showing backscattering coefficients for 2-3 and 3-4 April.

*The authors argue, it has to do with the chemical composition of the dust particles, and then with the imaginary part? But my question is continuously: If we use an alternative shape model what would then be the result? Gasteiger et al., and modelling groups in Finland introduced very different shape models. And found very different results in terms of dust optical properties.*

In Fig.5a we observe that lidar ratio at 532 nm decreases with height, while high depolarization ratio doesn't change noticeably. From this we conclude that we deal with "pure" dust, but properties of this dust change with height. It can be particle size, shape, composition (and so imaginary part). At current stage we are not able to specify the mechanism. At a moment we have no rigid model that can be used for simulation of highly variable real dust particles. The only model available today (allowing to perform computations for reasonable time) is spheroids model, other models are not capable to treat the particles with radii as big as ~10 μm. And actually, spheroids work not so bad, as we can conclude, for example, from recent publication (Shin et al., ACP 18, 2018). However, if we talk about influence of the imaginary part, increase of Im will lead to decrease of backscattering for particles of any shape and lidar ratio should increase.

*And if we ignore all the giant dust particles in the complex data analysis, how large can be the damage?*

In observations of extinction and backscattering, we don't make assumptions about particles size or shape. However we don't have enough information, to conclude how presence the giant particles will influence further analysis (for example volume retrieval). But influence of spectral dependence of the imaginary part for giant particles should be stronger.

*P7 l 210-220: The lidar ratios for dust (355 vs 532 nm) are so different (60-70sr vs 45-60sr), although the depolarization ratio shows (almost) pure dust conditions, this is a very surprizing result and in contradiction with the literature (Tesche et al, 2011, Gross et al., 2011). Any explanation for this? Maybe: : : it has to do again with missing coarse and giant, irregularly shaped dust particles? Because AE(S) = - AE(bsc), why is backscatter Angstroem exponent so pronounced, why is that so negative? Is everything ok with the backscatter retrieval (calibration in the reference height)?*

Our explanation is spectral dependence of the imaginary part. Increase of Im at 355 nm today is well proved by dust sample analysis. Typical increase of Im of dust is about 0.002 – 0.003, when wavelength decreases from 532 to 355 nm, and modeled ratio S355/S532 agrees with observations. The increase of lidar ratio at 355 should be for any particle shape.

*explanation would be: smoke is responsible for the strong lidar ratio wavelength dependence, but then the depolarization ratio should indicate that.*

This is what we try to show in manuscript: the smoke presence can't explain such behavior. But increase of imaginary part – can.

*P9 L266-267: There are also smoke lidar ratios from the SAMUM Cabo Verde campaign (Tesche, Tellus, 2011, second paper).*

Added

*P11 L325-326: The lidar ratio depends on complex refractive index and size distribution, BUT ALSO ON SHAPE of the particles.*

Definitely!

*Again the conclusion from my side: Are all the simulations and retrieval products trustworthy and reflect the reality when a spheroidal shape model is needed and used? Can we trust the main findings and conclusions?*

Comparison of spheroids model with observations demonstrate that it reasonably reproduces observed lidar ratios, for example (Shin et al., ACP 18, 2018). So we think that findings presented are trustable. But definitely, spheroids model is not the ultimate solution and the models, accounting for particles with sharp faces should be considered.

*And again: What about the impact of missing giant, irregularly shaped particles? They are there over-near source regions!*

We have not enough information about properties and content of such particles for conclusions. But presence of giant particles will probably make influence of spectral dependence of Im even stronger.

*P11 L355-356: observed low values of AE(bsc) cannot be produced without accounting*
*for a spectral dependence of the imaginary part:. Yes, for the case of the assumed*
*spheroidal shape model and size distribution up to about 10_m radius.*
*AERONET is used for comparison. But here (photometer) also a spheroidal shape*
*model is needed for dust particles. So, no independence between lidar and photometer*
*products.*

In the case of AERONET the particle shape is not so critical, because wide range of angles is considered. And again, observed lidar parameters agree reasonably with AERONET results. So we think that AERONET is the most reliable source of information at a moment. Just want to mention, that AERONET was criticized for long time for too low values of imaginary part in UV. But recent sample measurements ( Di Biagio et al., 2019) demonstrate that previously used values of imaginary part (and real as well) of dust at 355 nm were strongly overestimated and actually AERONET provides reasonable values.

*P12 L368-369: the authors write: :again the observed S355/S532 ratio*
*and AE(bsc) can be explained by the spectral dependence of the imaginary part of*
*CRI. My answer again: Yes, for spheroidal particles. So again the reader is left with*
*the question, and what is now the true impact of the imaginary part for irregularly*
*shaped particles for which we do not have a model?*

As mentioned, increase of the imaginary part will always lead to decrease of backscattering (radiation is absorbed in particle) and to increase of lidar ratio. But the absolute values of lidar ratios, will definitely depend on particle shape (this is why we use spheroids instead spheres). We add corresponding phrase in Conclusion of revised manuscript.

*P12 L371: S355/S532 is 1.5 !!! and the depol ratio >0.3 indicates pure dust! Such*
*a result I have never seen in the literature, and thus must have to do with the field*
*site (and probably to the omnipresence of very large dust particles). I have no other*
*explanation. Disregarding, whether my comment makes sense or not, the findings*
*in this paper seem to be specific for Senegal, or maybe regions in the vicinity of dust*
*source regions. That should be clearly written.*

Yes, such high S355/S532 were observed only during strong dust episodes. Moreover, in many cases we observed "traditional" lidar ratios. Actually the goal of this manuscript was to show, that ratio S355/S532 >1, when imaginary part in UV (provided by AERONET) is high, and S355/S532 =1 when Im is low. We observe good correlation between AERONET and lidar data. These instruments use different principles and are independent. So there should be some true behind our findings!

*P13 L418: The authors write: The relative humidity varied from episode to episode.*
*: : :. To my knowledge, different RH conditions usually show different air masses and*
*different aerosol properties from different sources. At least it seems to be a difficult task*
*to correlate RH with found aerosol properties, and to draw trustworthy conclusions on*
*the hygroscopic (water uptake) properties.*
*P14, L447-448 Ok, this is said now but that comes very late.*

In revised manuscript we put it earlier.

*P14 : When modelling lidar ratios for smoke, do you have proper smoke particle size distributions as input?*

We used particle parameters from GEOS-5 model. Smoke parameters in different regions are too variable to be described by just one model. In our study we tried to compare model and our observations, to see if the model reproduces reasonably the lidar observed parameters.

*P15 L476-478: Also the size distribution changes a bit when the relative humidity increases. Is that considered in the simulations?*

Yes, PSD changed with RH in modeling.

*P17 after L545-550: At, at the end of the summary and conclusion section, one could mentioned as an outlook, that such studies as presented here should be repeated at many different places around the world, e.g. in the Middle East, Central and East Asia, Australia, and North Amerika in order to improve our knowledge on real-world aerosol optical properties needed in optical modelling and climate relevant atmospheric modelling.*

Yes, we absolutely agree. Corresponding comment is added to conclusion.

*My final summarizing comment: Disregarding the criticisms made here, my overall impression is: The discussion and sensitivity studies are nice and very helpful to interpret sophisticated multiple wavelength polarization Raman or HSRL lidar observations of complex dust and dust-smoke mixtures. The authors are experts and contribute with an important contribution to the field of dust atmospheric studies and interpretation in terms of dust size distribution, shape properties, and chemical composition. However, the discussion should be more sensitive to all the shortcomings (unknown shape, ignored giant particles). The more clearly the short comings are presented the more exciting the story and the probability to trigger further studies. The field is interesting, and the more questions are left at the end, the more interesting is this research field for the next generation of researches. To define the right questions is often better then to present some (questionable) answers: : :*

We agree with reviewer that dust is a complicated object and our study tries to figure out the main factors (for example spectral dependence of Im), influencing the observed values. But definitely more studies in this field are needed.

**Figures:**

*Fig.1: Please skip the symbols, just show different colored thin lines. It is hard to see anything, except the ups and downs in the curves.*

The AERONET data points are scattered, so it is not good to use lines. In revised figure we decreased the symbol size. Now it should be better.

*Fig. 5a: The lidar ratios of 70 sr at 355 nm and 50 sr at 532 nm clearly shows the strong impact of smoke to my opinion. The extinction profiles and the depolarization ratio > 0.3 indicate pure dust! I am very much confused.*

The depolarization is too high for smoke. The goal of this manuscript is to show, that such high lidar ratios are explained by increase of the imaginary part.

*Fig.5b: The same here. And the modelling results are at all even more extreme and thus totally unrealistic to my opinion. with the spheroidal model as Input almost 40 sr at 532 nm and 70 sr at 355 nm for pure dust conditions! Has this*

*something to do with any dust reality? And AE(S) of 1.2 or and AE(Bsc) of -1.2 from the model?*

The model GEOS-5 uses Im=0.007 at 355 nm, which is quite reasonable (though near the up boarder). Computations are done for randomly oriented ellipsoids (results are very close to spheroids). The lidar ratio S355=70 sr predicted by the model was observed in our measurements during strong dust episodes. And when we have mixture of smoke and dust, the AE(ext) will be positive while AE(back) can be negative.

*Fig 6: Here one could discuss AE(S)=AE(ext)-AE(bsc) when showing AE(ext) and AE(bsc). AE(ext) at least shows what we know from AERONET and the many dust lidar observations. But the backscatter wavelength dependence remains unique and seems to be quite special for this part of the world, whatever the reason may be.*

We agree the observations at different locations are needed.

*Fig. 8: Again, nicely measured 355 and 532 nm extinction profiles (a), and (surprisingly) 'well-known' lidar ratios in the upper part of the dust layer, and very trustworthy depol ratios (b) and trustworthy AE(ext) profile (b), and even AE(bsc) and thus AE(S) in the upper part of the layer, the same for (c) and (d). This is what we know from the literature and all the SAMUM and SALTRACE findings for Saharan dust and aged outflow dust transported to the west.*

This is the message of the manuscript! We can observe "traditional" lidar ratios and enhanced ones. And it correlates with enhanced imaginary part provided by AERONET. This is why we think that enhanced absorption is the reason.

*And then again this huge contrast: AGAIN these strange model results. Lidar ratios up to almost 70sr at 355 nm and down to 40sr at 532 nm, and at the same time, these very reasonable lidar measurements. The authors seem to provide me (voluntary) with the wappons I need to blame the modelling effort and point on the used shape model as the main source for these errors. What shall we believe at the end, from all the conclusions the author derive, when we see this? Fig.8 is an excellent figure to corroborate my opinion: I do not believe in any of the result!*

The model results are not strange at all. As soon as we consider spectral dependence of the imaginary part we will have enhanced values S355 for any particle shape. But exact values of lidar ratios will definitely depend on the particle shape. But from comparison of model and observations we conclude that model works actually not so bad.

*Fig 10: The depolarization ratio shows no change in the dust conditions over the plotted one month period and is so high (PURE DUST), but the lidar ratio is sensitively changing: and again probably solely dependent on the backscatter coefficient which is a strong function of refractive index and particle shape! So, what to do? Ok there is a correlation with the refractive index from AERONET. But AERONET also assumes a spheroidal model to produce dust-related results. So, what can we believe at the end.*

Yes, this is what we wanted to show. That depolarization is quite stable, but ratio S355/S532 varies. Because backscattering is very sensitive to change of the imaginary part. The real part can also influence the variations, but, being spectrally independent, it should influence both S532 and S355. Besides, AERONET does not report significant variation of the real part. Particle shape variation can definitely influence the lidar ratio. But then we would expect variation of depolarization ratio also, which does not happen. And finally, we see correlation of S355/S532 changes with Im provided by AERONET. The instruments are independent, so it can not be just coincidence.

*Fig 12: Similar curves , good! But what does it help?*

Yes, it shows correlation between lidar and AERONET very clear.

*Fig 13: Size distribution from AERONET and from lidar (at what height?).*

Within 2-3 km range. This is in the text and we added this value to the figure capture.

*Differences are visible but both size distributions stop at 10 microns. What about bigger particles when measuring very close to the source region: Is it possible that the used size distribution in all model runs underestimate the impact of the giant particles?*

We don't have enough information about giant particles content to estimate their influence. But the influence of the imaginary part probably will be even higher.

Fig. 14: I do not believe the Imag curve (stars) as long as we have another errors source (the spheroidal shape model). AE(Bsc) = - AE(S) can be used her in the discussion because AE(ext) is almost zero.

Drop of AE(bsk) with Im increase will be for any particle shape. Comparison the values of lidar ratios with predictions of spheroids model shows actually this model works not so bad (Shin et al., ACP 18, 2018).

*Fig 17-Fig 19: The RH study and discussion makes the paper quite long, and all the efforts are again quite speculative: : : Fig 19 is not convincing.*

We agree that in Fig.19 there are too many assumptions. Figure is removed from revised manuscript. But the rest of results we would prefer to keep, because these demonstrate the properties of dust – smoke mixtures for different RH.

---

## Author Comment (AC2) · 22 Apr 2020

We are very grateful to Referee for job, he has done edditing our maniscript. We incorporated his recommendations in the revised manuscript

---

## Author Comment (AC3) · 22 Apr 2020

**Response to comments of Reviewer #4**

First of all, we would like to thank the Reviewer for careful reading the manuscript and for useful suggestions.

**MINOR REVISIONS**

*I have a personal curiosity related to the paper: The authors have a lot of works on retriving aerosol microphysical properties from 3b+2a lidar measurements, even for non-spherical particles. Here, authors have the measurements for the retrieval. I would like to know why authors have decided not to do the 3b+2a inversion to retrieve aerosol refractive index. I have also follow your last papers in retrievals of aerosol microphysical properties from space-borne simulations and I would like to know if your new results can have an impact in space retrievals.*

We think that consideration of variation of the imaginary part (and its spectral dependence) is important for both dust and biomass burning products and should be included in simulation of space based lidars simulation. Corresponding references are added to Conclusion.

*In the discussion of changes in lidar ratios for smoke with relative humidity, please take into account that they do not only depend on refractive index. Also is important the possible changes in size distribution.*

The model considers variation of PSD with RH, so it is included.

*Line 51-55: Please, note that spectral dependence in lidar ratio have been demonstrate useful to estimate the range of refractive index for non-spherical particles, although assuming no spectral dependence in CRI*

Yes, preliminary estimation of range of refractive index (RI) variation is important. Still we need to make next step and include spectral variation of RI in inversion.

*Line 77: Please, define variables. What is Im440 and Im355.*

Done

Line 110: Why measurements are acquired at 47 degrees angle to horizon?

We measured through the window in the room, so it was the largest possible angle.

*Equation 1: Please, give a proper reference*

We derived it ourselves. It is quite straighforward.

Line 163: I thing there is a type in 'recalculated'

We think that "recalculated" looks correct in the context used…

*Line 204: ' The backcatter Angström exponent Ab, in contrast with Az is sensitive to the spectral dependence of the imaginary part of CRI'. Please clarify and provide references. Actually, Aa also depends on imaginary refractive index.*

Here we paste Figure from (Veselovskii et al., 2010). Corresponding reference is added to the text.

[Figure]

**Figure 6.** The particle backscattering (circles) and extinction (stars) coefficients at 355 nm wavelength as a function of the imaginary part of the refractive index. Calculations for spheres (open symbols) and spheroids (solid symbols) were performed for the $PSD_{20}$ with $m_R = 1.55$, $N_f = 100$ $cm^{-3}$.

Yes, backscattering depends also on the real part (Re), but Re for dust doesn't show significant spectral dependence.

*Line 245: Please explain futher why you assume 35% and 7% for dust and depolarization ratio or provide appropiate references.*

35% is the highest depolarization ratio we observed for pure dust. The depolarization ratio of different types of smoke can vary significantly. 7% is the lowest value we observed in elevated smoke layers during SHADOW. We should mention, that depolarization of smoke is much lower than that of dust, thus the choice of exact value of smoke depolarization does not influence significantly the results. Corresponding comment and reference is added to manuscript.

*Line 258: It is not clear why 25 sr is unrailstic lidar ratio. Please, provide references.*

We added reference to Burton et al., 2012.

Line 267: Why smoke lidar ration should increase with RH ?

This is combined effect of size increasing and the real part decrease.

*Line 276: the statement 'dust became less absorbing in the UV' is unclear. Are you referring to imaginary refractive index or to single scattering albedo?*

Dust becomes less absorbing due to decrease of the imaginary part, which in turn leads to increase of SSA. Effective radius (as follows from AERONET) didn't change significantly during this period.

*Lines 338-353: Authors give a description of the 3b+3a lidar inversion. That should come earlier becase previously in Figure 13 you show size distribution from 3b+2a inversion.*

We actually don't describe the inversion, just provide the reference. It was described for many times previously.

*Lines 333-337: It is not clear to me how you make the simulations.*

We computed extinction and backscattering coefficients for different Im using spheroids model. From these data the lidar ratios and Angstrom exponents were obtained. To estimate influence of spectrally dependent Im, we used $\beta$ and $\alpha$ computed for different Im at 355 and 532 nm.

Table 1: How you estimated uncertainties?

For lidar ratios we considered only statistical errors. For RH, we took upper and lower limits of lidar derived water vapor mixing ratio, basing on uncertainty of calibration. For these values RH was calsulated.

*Lines 465-467: Please, note the limitations in PSD variability with relative humidity in MERRA-2.*

Yes, PSD of smoke particles can vary with RH differently for different types of smoke. A single model can't describe all variability of smoke particles, so one of the goals of this work was comparison of MERRA-2 predictions with observations. We conclude that the model reproduce the general tendency of increase of the lidar ratio with RH, if the initial values of the imaginary part of dry particles are chosen correctly.

*Conclusion section: 'Our study shows the impact of aerosol spectral absorption variation on the lidar-derived aerosol properties'. Do you refer to any aerosol type? I think you want to say dust and smoke aerosol.*

Yes, this is for dust – smoke mixture. We corrected it in Conclusion.

Figure 7a. There is a blue line missing.

Contribution if Sea Salt is just very low and can't bee seen on a figure.

---

## Author Comment (AC4) · 22 Apr 2020

**Response to Referee #3**

First of all, we would like to thank the Referee for careful reading the manuscript and for useful suggestions.

Answering the Referee comments:

*Fig.8a and Page 9, Lines 283-285: "Thus, we can assume that increase of the imaginary part in UV in the first layer is more significant, than in the second one".*
*I noticed from this figure that the Angstrom exponent (both backscatter and extinction related) increases towards higher altitudes, which coincides with a slight decrease of depolarization ratio and a coincidence of the S355 and S532. Could these variations point towards the dominance of smaller dust particles higher in the layer? From laboratory studies we know that smaller dust particles present lower depolarization ratio values (i.e. Järvinen et al. 2016; Sakai et al., 2010), while also the larger S532 values lower in the layer could be attributed to the increased "sensitivity" this wavelength should have to the presence of larger particles. Why is the dominance of smaller dust particles should be excluded here?*

Slight increase of the extinction Angstrome exponent (EAE) with height can definitely indicate that particles become smaller, which agrees with decrease of depolarization ratio. But in previous dust episodes for the same EAE and depolarization ratio, we had S355>S532, so we think that change of particle size only, without decrease of the imaginary part in UV can not explain different values of S355/S532 in upper and lower layers. Besides, the AERONET data demonstrate that column imaginary part in UV starts to decrease this day.

*Page 8, Line 245: "assuming 35% and 7% for dust and smoke depolarization ratio".*
*Please provide some references on choosing these values specifically. Did you use the same values also for the dust-smoke decomposition you perform for all the cases presented?*

35% is the highest depolarization ratio we observed for pure dust. The depolarization ratio of different types of smoke can vary significantly. 7% is the lowest value we observed in elevated smoke layers during SHADOW. We should mention, that depolarization of smoke is much lower than that of dust, thus the choice of exact value of smoke depolarization does not influence significantly the results. Corresponding comment and reference is added to manuscript.

*Also Page 14, Line 429: "In principle, we can estimate S532s using Eq.5, because the ratio β532s)/β532 is available". Here the 532s depends on the selected value of δ532s. Can you provide an estimation of the uncertainties of this approach? What could be the effect on the resulting S532s values?*

Yes, choice of smoke depolarization ratio introduces uncertainty in estimation. However, due to large difference of dust and smoke depolarization, the choice of exact value of smoke depolarization does not influence significantly the results, especially at the heights where smoke is predominant. Another source of uncertainty is choice of the dust lidar ratio, which can provide uncertainty about 5 sr. The estimations of smoke lidar ratio are qualitative, so in final plot we show values of lidar ratio for the smoke – dust mixture only.

*Table 1: Could you add to the table the height intervals chosen for the analysis of the smoke layers and also an estimation of their lifetime?*

Unfortunately we can not do it, because in many cases the smoke occurs at all heights, so it is not easy to define the height range. In Table 1 we provide results for the heights, where the ratio $\beta_s/\beta$ is maximal, which is usually near the top of the smoke layer. The same is for the smoke layers life time. During December – January, the smoke occurred almost permanently, so it is difficult to define the beginning and the end of smoke episode. Besides, we performed measurements in the night time only.

*Could any differences in the smoke properties be related to the age of smoke particles*?

The smoke aging may contribute to observations; this is why we say that presented results should be taken as semi-qualitative only. Still results show the general tendency of the lidar ratio increase with RH.

*Page 4, Line 113: for the range resolution of particle extinction coefficient it is not clear to me which height intervals are selected. Do you mean 50 m up to 1000m and 125 m from 1000m to 7000m?*

Yes, range resolution is 50 m up to 1000 m and then it gradually increases up to 125 m at 7000 m range. Please, keep in mind, that we performed observations at 47 deg to horizon, so to get height resolution it should be multiplied by factor 0.73.

*Page 7, Line 217: the authors probably mean "hydrophobic".*

Yes, sorry. Computer corrected it automatically. Changed.

*Page 11, Line 344: "spectrally independent refractive index". Please provide the selected values for this analysis.*

In inversion we consider both real and imaginary parts of CRI in a wide range of values: 0<Im<0.02; 1.4<Re<1.65. This procedure was described in details in our previous publications, and we didn't repeat it here, because we did not focus on inversion in this study.

*Page 12, Line 369: "so variation of the imaginary" add "part of the refractive index".*

Corrected

Section 3.2: Please provide the spatial-temporal evolution of backscatter coefficient, water vapor and particle depolarization for these cases also.

Optical depth of aerosol layer was high for 13-14 and 23-24 April, so to get reliable reference point for backscattering calculation, several profiles should be averaged. Hence, we can't provide spatio-temporal distributions for backscattering as in Fig.3. We can provide range corrected lidar signal, or backscattering calculated by Klett. However, dust layer didn't change much during the night, so we think that no need to add extra figure. Besides, manuscript is already overloaded with figures.

---

## Author Response (AR2)

In revised manuscript we added explanation of weak dependence of extinction on the imaginary part.

Ln 71. The dust backscattering coefficient, in contrast to the extinction coefficient, is sensitive to the imaginary part of CRI (Perrone et al., 2004; Veselovskii et al., 2010; Gasteiger et al., 2011). Recall, that the particle extinction is the sum of absorption and scattering, and increase of absorption is accompanied by decrease of scattering, leading to weak dependence of the extinction on the imaginary part (Veselovskii et al. 2010, Fig.6).